# A Generative Model of Symmetry Transformations

**James Urquhart Allingham**
University of Cambridge
jua23@cam.ac.uk

**Bruno Kacper Mlodozeniec**
University of Cambridge
MPI for Intelligent Systems, Tübingen
bkm28@cam.ac.uk

**Shreyas Padhy**
University of Cambridge
sp2058@cam.ac.uk

**Javier Antorán**
University of Cambridge
Ångstrom AI
ja666@cam.ac.uk

**David Krueger**
University of Cambridge
david.scott.krueger@gmail.com

**Richard E. Turner**
University of Cambridge
ret26@cam.ac.uk

**Eric Nalisnick**
University of Amsterdam
e.t.nalisnick@uva.nl

**José Miguel Hernández-Lobato**
University of Cambridge
jmh233@cam.ac.uk

## Abstract

Correctly capturing the symmetry transformations of data can lead to efficient models with strong generalization capabilities, though methods incorporating symmetries often require prior knowledge. While recent advancements have been made in learning those symmetries directly from the dataset, most of this work has focused on the discriminative setting. In this paper, we take inspiration from group theoretic ideas to construct a generative model that explicitly aims to capture the data's *approximate* symmetries. This results in a model that, given a prespecified broad set of possible symmetries, learns to what extent, if at all, those symmetries are actually present. Our model can be seen as a generative process for data augmentation. We provide a simple algorithm for learning our generative model and empirically demonstrate its ability to capture symmetries under affine and color transformations, in an interpretable way. Combining our symmetry model with standard generative models results in higher marginal test-log-likelihoods and improved data efficiency.

## 1 Introduction

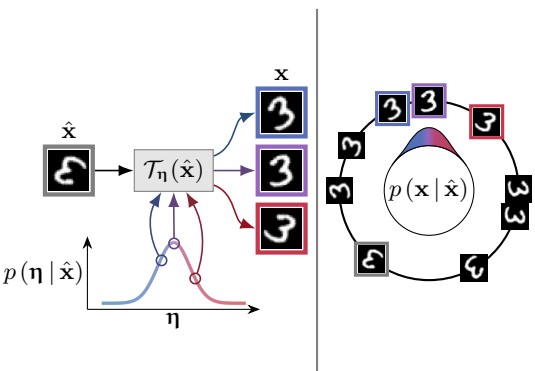

Figure 1: **Left:** An example of a symmetry-aware generative process that we aim to model in this paper. A *prototype* $\hat{\mathbf{x}}$ (□) is transformed by $\mathcal{T}_{\boldsymbol{\eta}}$ into an observation $\mathbf{x}$ (□, □, □). The transformation—e.g., rotation—is parameterized by $\boldsymbol{\eta}$—e.g., an angle. **Right:** The corresponding orbit—i.e., the set of all possible instances of $\mathbf{x}$ that can result from applying $\mathcal{T}_{\boldsymbol{\eta}}$—with a few elements shown. Under this generative process, the prototype is an arbitrary orbit element. Each element in the orbit has a probability $p(\mathbf{x} \mid \hat{\mathbf{x}})$ induced by $p(\boldsymbol{\eta} \mid \hat{\mathbf{x}})$. E.g., for handwritten '3's, we expect digits in an upright orientation with some rotation around, say $\pm 40°$, corresponding to natural variations in handwriting.

Many physical phenomena exhibit symmetries; for example, many of the observable galaxies in the night sky share similar characteristics when accounting for their different rotations, velocities, and sizes. Hence, if we are to represent the world with generative models, they can be made more faithful and data-efficient by incorporating notions of symmetry. This has been well-understood for discriminative models for decades. Incorporating inductive biases such as invariance or equivariance to symmetry transformations dates back (at least) to ConvNets, which incorporate translation symmetries [LeCun et al., 1989]—and can be extended to reflection and rotation [Cohen and Welling, 2016]—and more recently, transformers, with permutation symmetries [Lee et al., 2019].

In many cases, it is not known *a priori* which symmetries are present in the data. Learning symmetries in discriminative modeling is an active field of research [Nalisnick and Smyth, 2018, van der Wilk et al., 2018, Benton et al., 2020, Schwöbel et al., 2021, van der Ouderaa and van der Wilk, 2022, Rommel et al., 2022, Romero and Lohit, 2022, Immer et al., 2022, 2023, Miao et al., 2023, Mlodozeniec et al., 2023]. However, in these works—which focus on invariant discriminative models—the label is often assumed to be invariant, and thus, the symmetry information can be *removed* rather than explicitly modeled. On the other hand, a generative model *must* capture the factors of variation corresponding to the symmetry transformations of the data. Doing so can provide benefits such as better representation learning—by disentangling symmetry from other latent variables [Antorán and Miguel, 2019]—and data efficiency—due to compactly encoding of factor(s) of variation corresponding to symmetries. Furthermore, learning about underlying symmetries in data could be used for scientific discovery.

We propose a generative model that explicitly encodes the (partial) symmetries in the data. Here, we are primarily interested in using this model to inspect the distribution over naturally occurring transformations for a given example $\mathbf{x}$, and resample new "naturally" augmented versions of the example. Our contributions are

1. We propose a Symmetry-aware Generative Model (SGM). The SGM's latent representation is separated into an invariant component $\hat{\mathbf{x}}$ and an equivariant component $\boldsymbol{\eta}$. The latter, $\boldsymbol{\eta}$, captures the symmetries in the data, while $\hat{\mathbf{x}}$ captures none. We recover $\mathbf{x}$ by applying a parameterised transformation, $\mathbf{x} = \mathcal{T}_{\boldsymbol{\eta}}(\hat{\mathbf{x}})$. We call $\hat{\mathbf{x}}$ a *prototype* since each $\hat{\mathbf{x}}$ can produce arbitrarily transformed observations; see Figure 1.

2. We propose a two-stage algorithm for learning our SGM: first learning $\hat{\mathbf{x}}$ using a self-supervised approach and then learning $\boldsymbol{\eta}$ via maximum likelihood. Importantly, this does not require modeling the distribution of prototypes $p(\hat{\mathbf{x}})$, allowing the procedure to remain tractable even for complex data.

3. We verify experimentally that our SGM completely captures affine and color symmetries. A VAE's marginal test-log-likelihood can improved by using our SGM to incorporate symmetries. Additionally, unlike a standard VAE, explicitly modeling symmetries makes our VAE-SGM hybrid robust to deleting half of the dataset.

**Notation.** We use $a$, $\boldsymbol{a}$, and $\boldsymbol{A}$ (i.e., lower, bold lower, and bold upper case) for scalars, vectors, and matrices, respectively. We distinguish between random variables such as $\mathbf{x}$, $\boldsymbol{\eta}$, $\mathbf{A}$, and their realizations $\boldsymbol{x}$, $\boldsymbol{\eta}$, $\boldsymbol{A}$. Thus, for continuous $\mathbf{a}$, $p(\mathbf{a})$ is a PDF that returns a density $p(\mathbf{a} = \boldsymbol{a}) = p(\boldsymbol{a})$. We use $\circ$ to represent function composition, e.g., $f_1 \circ f_2$.

## 2 Symmetry-aware Generative Model (SGM)

Consider a dataset of observations $\{\boldsymbol{x}_n\}_{n=1}^N$ on a space $\mathcal{X}$, and a collection $\{\mathcal{T}_{\boldsymbol{\eta}}\}$ of transformations $\mathcal{T}_{\boldsymbol{\eta}} : \mathcal{X} \to \mathcal{X}$ parameterised by transformation parameters $\boldsymbol{\eta} \in \mathcal{H} \subseteq \mathbb{R}^{d_\eta}$. We assume $\{\mathcal{T}_{\boldsymbol{\eta}}\}_{\boldsymbol{\eta} \in \mathcal{H}}$ (abbreviated $\{\mathcal{T}_{\boldsymbol{\eta}}\}$) form a group. Loosely, our aim is to model the distribution over transformations present in the data. To do so, we model the distribution $p(\mathbf{x})$ by decomposing it into two disparate parts: **(1)** a distribution over prototypes and **(2)** a distribution over parameters controlling transformations to be applied to a prototype. Concretely, we specify our generative model as follows (also depicted in Figure 2):

$$\hat{\mathbf{x}} \sim p(\hat{\mathbf{x}}), \tag{1}$$

$$\boldsymbol{\eta} \sim p_{\boldsymbol{\psi}}(\boldsymbol{\eta} \,|\, \hat{\mathbf{x}}), \tag{2}$$

$$\mathbf{x} = \mathcal{T}_{\boldsymbol{\eta}}(\hat{\mathbf{x}}). \tag{3}$$

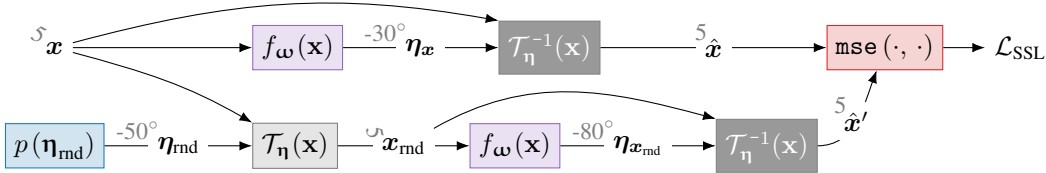

Figure 4: Self-supervised symmetry learning. We encourage $f_{\boldsymbol{\omega}}(\mathbf{x})$ to be equivariant by mapping $\boldsymbol{x}$ and a randomly transformed $\boldsymbol{x}$ to the same $\hat{\boldsymbol{x}}$. Gray text shows examples for each variable in the graph. Note that $\hat{\boldsymbol{x}}$ and $\boldsymbol{x}_{\mathrm{rnd}}$ may not appear in the dataset; see Figure 1.

That is, the SGM assumes that each observation $\mathbf{x}$ is generated by applying a transformation $\mathcal{T}_{\boldsymbol{\eta}}$—parameterized by a latent variable $\boldsymbol{\eta}$—to a latent prototype $\hat{\mathbf{x}}$. Since $\hat{\mathbf{x}}$, by assumption, contains no information about the symmetries in the data, $p_{\boldsymbol{\psi}}(\boldsymbol{\eta} \mid \hat{\mathbf{x}})$ must model the distribution over the transformations $\mathcal{T}_{\boldsymbol{\eta}}$ present in the data.

**Motivation.** Why would we expect specifying $p(\mathbf{x})$ in this way to be useful? Firstly, our SGM allows us to query a distribution over naturally occurring transformations $p_{\boldsymbol{\psi}}(\boldsymbol{\eta} \mid \hat{\mathbf{x}} = \mathcal{T}_{\boldsymbol{\eta}}^{-1}(\boldsymbol{x}))$ for any input $\boldsymbol{x}$, given the matching prototype $\hat{\boldsymbol{x}} := \mathcal{T}_{\boldsymbol{\eta}}^{-1}(\boldsymbol{x})$. Secondly, we expect our SGM to align with the true physical process of generating the data for many interesting datasets. As an illustrative example, when a person writes a digit, they first decide what kind of digit to write—e.g., the prototype could be an upright '3'—but when they put pen to paper, the digit

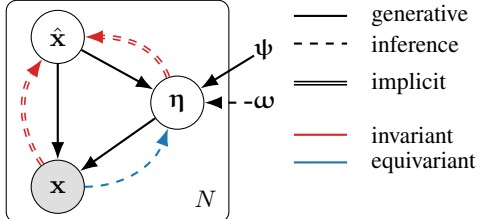

Figure 2: SGM graphical model. The *implicit* edges denote that $\hat{\mathbf{x}}$ is fully specified by $\boldsymbol{\eta}$ and $\mathbf{x}$—since $\hat{\mathbf{x}} = \mathcal{T}_{\boldsymbol{\eta}}^{-1}(\mathbf{x})$—and thus only $\boldsymbol{\eta}$ needs to be inferred given and observation $\mathbf{x}$.

they pictured is transformed due to various factors governing their handwriting[1]. Similarly, when a photographer captures an object, the photo is also a function of latent factors of variation, such as lighting, the lens, camera shake, etc.

**What do we require of a prototype?** $\hat{\mathbf{x}}$ can informally be considered a canonical/reference example with no transformation applied to it. More precisely, we require that for any *orbit* of an element $\mathbf{x}$—defined as the set of elements in $\mathcal{X}$ which $\mathbf{x}$ can be mapped to by a transformation in $\{\mathcal{T}_{\boldsymbol{\eta}}\}$—there is exactly one prototype in the orbit. Figure 1 depicts an example orbit—a set $\{\blacksquare, \blacksquare, \blacksquare, ...\}$ of all rotated variants of a '3'—with a unique prototype.

**Why do we want a group?** Having the transformations $\{\mathcal{T}_{\boldsymbol{\eta}}\}$ be a group simplifies things, since $\{\mathcal{T}_{\boldsymbol{\eta}}\}$ will then naturally partition the space $\mathcal{X}$ into (disjoint) orbits. Within each orbit, every element can be transformed into one another with a transformation in $\{\mathcal{T}_{\boldsymbol{\eta}}\}$. As an example of such a partition, if our collection of transformations were horizontal shifts $\mathcal{T}_{\boldsymbol{\eta}} : \mathbf{x} \mapsto \mathbf{x} + (\eta, 0)$ acting on a point $\mathbf{x} \in \mathbb{R}^2$, then the different orbits will correspond to all points on a given horizontal line; see Figure 3. Therefore, if we have chosen a unique prototype for each orbit and $\{\mathcal{T}_{\boldsymbol{\eta}}\}$ forms a group, any two elements $\mathbf{x}, \mathbf{x}' \in \mathcal{X}$ will have the same prototype if and only if they can be transformed into one another.

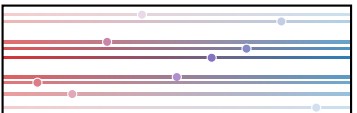

Figure 3: Orbits due to horizontal shift transformations. Each point $(x_1, x_2)$ is transformed via $\mathcal{T}_{\eta} : (x_1, x_2) \mapsto (x_1, x_2) + (\eta, 0)$. Thus, horizontal lines form disjoint orbits in which any point can be transformed into any other point on the same line but not on another line. For each line, we can choose an arbitrary prototype ( ● ) from which all other points on the line can be reached via $\mathcal{T}_{\eta}$.

In Section 2.1, we describe a method for learning a transformation inference function $f_{\boldsymbol{\omega}} : \mathcal{X} \to \mathcal{H}$, with parameters $\boldsymbol{\omega}$, that for $\mathbf{x} \in \mathcal{X}$ returns transformation parameters $\boldsymbol{\eta} \in \mathcal{H}$ as $\boldsymbol{\eta} = f_{\boldsymbol{\omega}}(\mathbf{x})$. These map $\mathbf{x}$ to a prototype $\hat{\mathbf{x}} := \mathcal{T}_{\boldsymbol{\eta}}^{-1}(\mathbf{x})$ that generates $\mathbf{x} := \mathcal{T}_{\boldsymbol{\eta}}(\hat{\mathbf{x}})$[2]. We then apply standard generative modeling tools to learn $p(\hat{\mathbf{x}}, \boldsymbol{\eta}) = p(\hat{\mathbf{x}}) \, p_{\boldsymbol{\psi}}(\boldsymbol{\eta} \mid \hat{\mathbf{x}})$ given the generated data pairs $\{\hat{\boldsymbol{x}}_n, \boldsymbol{\eta}_n\}_{n=1}^N$.

---

[1]Our SGM does not always perfectly match the data-generating process. E.g., a person is unlikely to "imagine" the same prototype for both a '6' or a '9'—which can often be transformed into one another with rotation.

[2]The transformation is not necessarily unique.

## 2.1 Learning

We now discuss learning for the two NNs required by our model, $f_{\boldsymbol{\omega}}(\mathbf{x})$ and $p_{\boldsymbol{\psi}}(\boldsymbol{\eta} \,|\, \hat{\mathbf{x}})$. In Appendix A, we connect our learning algorithm with MLL optimization using an ELBO.

**Transformation inference function.** For $\mathcal{T}_{\boldsymbol{\eta}}^{-1}$, with $\boldsymbol{\eta}$ given by $f_{\boldsymbol{\omega}}$, to map $\mathbf{x}$ to a prototype $\hat{\mathbf{x}}$, it must, by definition, map all elements in any given orbit to the same element in that orbit. In other words, the output of $\mathcal{T}_{f_{\boldsymbol{\omega}}(\boldsymbol{x})}^{-1}(\boldsymbol{x})$ should be *invariant* to transformations $\mathcal{T}_{\boldsymbol{\eta}'}$ of $\boldsymbol{x}$:

$$\mathcal{T}_{f_{\boldsymbol{\omega}}(\boldsymbol{x})}^{-1}(\boldsymbol{x}) = \mathcal{T}_{f_{\boldsymbol{\omega}}(\mathcal{T}_{\boldsymbol{\eta}'}(\boldsymbol{x}))}^{-1}\left(\mathcal{T}_{\boldsymbol{\eta}'}(\boldsymbol{x})\right), \;\; \forall \boldsymbol{\eta}' \in \mathcal{H}. \tag{4}$$

To learn such a function, we optimize for this property directly. To this end, we sample transformation parameters $\boldsymbol{\eta}_{\mathrm{rnd}}$ from some distribution over parameters $p(\boldsymbol{\eta}_{\mathrm{rnd}})$. This allows us to get random samples $\boldsymbol{x}_{\mathrm{rnd}} := \mathcal{T}_{\boldsymbol{\eta}_{\mathrm{rnd}}}(\boldsymbol{x}) \in \mathcal{X}$ in the orbit of any given element $\boldsymbol{x} \in \mathcal{X}$. Since we want full (i.e., strict) invariance, $p(\boldsymbol{\eta}_{\mathrm{rnd}})$ must have support on the entire orbit [van der Ouderaa and van der Wilk, 2022]. We then learn an equivariant via a self-supervised learning (SSL) scheme $f_{\boldsymbol{\omega}}$[3] inspired by methods like BYOL [Grill et al., 2020] and, more directly, BINCE [Dubois et al., 2021]. For example, we could use the objective illustrated in Figure 4:

$$\left\| \mathcal{T}_{f_{\boldsymbol{\omega}}(\boldsymbol{x}_{\mathrm{rnd}})}^{-1}(\boldsymbol{x}_{\mathrm{rnd}}) - \mathcal{T}_{f_{\boldsymbol{\omega}}(\boldsymbol{x})}^{-1}(\boldsymbol{x}) \right\|_2^2, \;\; \boldsymbol{x}_{\mathrm{rnd}} = \mathcal{T}_{\boldsymbol{\eta}_{\mathrm{rnd}}}(\boldsymbol{x}), \boldsymbol{\eta}_{\mathrm{rnd}} \sim p(\boldsymbol{\eta}_{\mathrm{rnd}}). \tag{5}$$

Our actual objective differs slightly. Since $\mathcal{T}_{\boldsymbol{\eta}'}(\boldsymbol{x}') = \mathcal{T}_{\boldsymbol{\eta}''}(\boldsymbol{x}'')$ implies $\boldsymbol{x}' = \mathcal{T}_{\boldsymbol{\eta}'}^{-1} \circ \mathcal{T}_{\boldsymbol{\eta}''}(\boldsymbol{x}'')$, we use

$$\left\| \mathcal{T}_{f_{\boldsymbol{\omega}}(\boldsymbol{x})} \circ \mathcal{T}_{f_{\boldsymbol{\omega}}(\boldsymbol{x}_{\mathrm{rnd}})}^{-1}(\boldsymbol{x}_{\mathrm{rnd}}) - \boldsymbol{x} \right\|_2^2. \tag{6}$$

This change allows us to reduce the number of small discretization errors introduced with each transformation application by replacing repeated transformations with a single composed transformation; see Section 3.1 for further discussion. Our SSL loss is given in line 1 of Algorithm 1.

**Generative model of transformations.** Once we have a prototype inference function, we simply learn $p_{\boldsymbol{\psi}}(\boldsymbol{\eta} \,|\, \hat{\mathbf{x}})$ by maximum likelihood on the created data pairs $\left\{ f_{\boldsymbol{\omega}}(\boldsymbol{x}_i), \mathcal{T}_{f_{\boldsymbol{\omega}}(\boldsymbol{x}_i)}^{-1}(\boldsymbol{x}_i) \right\}$. This is shown in line 8 of Algorithm 1. While we need to specify the kinds of symmetry transformations $\mathcal{T}_{\boldsymbol{\eta}}$ we expect to see in the data, by learning $p_{\boldsymbol{\psi}}(\boldsymbol{\eta} \,|\, \hat{\mathbf{x}})$ the model can learn the degree to which those transformations are present in the data. Thus, we can specify several potential symmetry transformations and learn that some are absent in the data. Furthermore, the required prior knowledge (the support of $p(\boldsymbol{\eta}_{\mathrm{rnd}})$) is small compared to what our SGM can learn (the shapes of the distributions for each of the *present* transformations).

Since we are primarily interested in using the model to **(a)** inspect the distribution over naturally occurring transformations for a given element $\boldsymbol{x}$, and **(b)** resample

---

**Algorithm 1** Learning

**Require:** initial parameters $\boldsymbol{\omega}_{\mathrm{init}}$ & $\boldsymbol{\psi}_{\mathrm{init}}$, dataset $\mathcal{D}$
1: **function** SSL_LOSS($\boldsymbol{x}, \boldsymbol{\omega}$)
2: $\quad \boldsymbol{\eta}_{\boldsymbol{x}} \leftarrow f_{\boldsymbol{\omega}}(\boldsymbol{x})$
3: $\quad \boldsymbol{\eta}_{\mathrm{rnd}} \sim p(\boldsymbol{\eta}_{\mathrm{rnd}})$
4: $\quad \boldsymbol{x}_{\mathrm{rnd}} \leftarrow \mathcal{T}_{\boldsymbol{\eta}_{\mathrm{rnd}}}(\boldsymbol{x})$
5: $\quad \boldsymbol{\eta}_{\boldsymbol{x}_{\mathrm{rnd}}} \leftarrow f_{\boldsymbol{\omega}}(\boldsymbol{x}_{\mathrm{rnd}})$
6: $\quad \boldsymbol{x}' \leftarrow \mathcal{T}_{\boldsymbol{\eta}_{\boldsymbol{x}}} \circ \mathcal{T}_{\boldsymbol{\eta}_{\boldsymbol{x}_{\mathrm{rnd}}}}^{-1}(\boldsymbol{x}_{\mathrm{rnd}})$
7: $\quad$ **output** mse($\boldsymbol{x}, \boldsymbol{x}'$)
8: **function** MLE_LOSS($\boldsymbol{x}, \boldsymbol{\omega}, \boldsymbol{\psi}$)
9: $\quad \boldsymbol{\eta}_{\boldsymbol{x}} \leftarrow f_{\boldsymbol{\omega}}(\boldsymbol{x})$
10: $\quad \hat{\boldsymbol{x}} \leftarrow \mathcal{T}_{\boldsymbol{\eta}_{\boldsymbol{x}}}^{-1}(\boldsymbol{x})$
11: $\quad$ **output** $-\log p_{\boldsymbol{\psi}}(\boldsymbol{\eta}_{\boldsymbol{x}} \,|\, \hat{\boldsymbol{x}})$
12: $\boldsymbol{\omega}, \boldsymbol{\psi} \leftarrow \boldsymbol{\omega}_{\mathrm{init}}, \boldsymbol{\psi}_{\mathrm{init}}$
13: **while** $\boldsymbol{\omega}$ not converged **do**
14: $\quad \boldsymbol{X} \leftarrow$ next_batch($\mathcal{D}$)
15: $\quad$ update $\boldsymbol{\omega}$ with $\nabla_{\boldsymbol{\omega}} \frac{1}{B} \sum_{b=1}^{B}$ SSL_LOSS($\boldsymbol{X}_b, \boldsymbol{\omega}$)
16: **while** $\boldsymbol{\psi}$ not converged **do**
17: $\quad \boldsymbol{X} \leftarrow$ next_batch($\mathcal{D}$)
18: $\quad$ update $\boldsymbol{\psi}$ with $\nabla_{\boldsymbol{\psi}} \frac{1}{B} \sum_{b}$ MLE_LOSS($\boldsymbol{X}_b, \boldsymbol{\omega}, \boldsymbol{\psi}$)
19: **output** $\boldsymbol{\omega}, \boldsymbol{\psi}$

---

new "naturally" augmented versions of the element, we *do not* need to learn $p(\hat{\mathbf{x}})$. We can do **(a)** by querying $p(\boldsymbol{\eta} \,|\, \hat{\mathbf{x}} = \hat{\boldsymbol{x}})$ for $\hat{\boldsymbol{x}} := \mathcal{T}_{f_{\boldsymbol{\eta}}(\boldsymbol{x})}^{-1}(\boldsymbol{x})$, and we can do **(b)** by sampling $\boldsymbol{\eta} \sim p(\boldsymbol{\eta} \,|\, \hat{\boldsymbol{x}})$ and transforming the $\hat{\boldsymbol{x}}$ to get $\boldsymbol{x} := \mathcal{T}_{\boldsymbol{\eta}}(\hat{\boldsymbol{x}})$. Of course, if one wanted to sample new prototypes, one could fit $p_{\boldsymbol{\theta}}(\hat{\boldsymbol{x}})$ using, e.g., a VAE. Not learning $p(\hat{\boldsymbol{x}})$ greatly simplifies training for complicated datasets that would otherwise require a large generative model, an observation made by Dubois et al. [2021].

---

[3]If $f_{\boldsymbol{\omega}}$ is equivariant by *construction*, our SSL scheme is unnecessary. Alas, such constructions are unknown for many transformations, like those in this paper. Thus, we provide a *general* method for learning equivariances.

(a) Distribution for η given **x** and $\hat{x}$.

| $x$ | $\hat{x}$ | $p(\eta \mid \boldsymbol{x},\, \hat{\boldsymbol{x}})$ |
|---|---|---|
| 8 | 8 | $0.5 \cdot \delta(\eta - 0°) + 0.5 \cdot \delta(\eta - 180°)$ |
| ၆ | 8 | $0.5 \cdot \delta(\eta - 30°) + 0.5 \cdot \delta(\eta + 150°)$ |
| 8 | 8 | $0.5 \cdot \delta(\eta + 30°) + 0.5 \cdot \delta(\eta - 150°)$ |

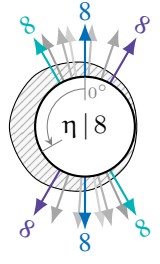

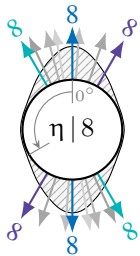

(b) Simple $p_\psi(\eta \mid \hat{x})$  (c) Flexible $p_\psi(\eta \mid \hat{x})$

Figure 5: Idealised examples of simple and flexible learned distributions over angles $p_\psi(\eta \mid \hat{x})$— ▨ —given the true distribution $p(\eta \mid \hat{x}) = \sum_{\mathbf{x} \in \{ ၆,\ldots,8,\ldots,8 \}} p(\eta \mid \mathbf{x},\, \hat{x})$— ▲▲▲ .

## 3 Practical Considerations and Further Motivations

Training our SGM, while simple, has potential pitfalls in practice. We discuss the key considerations in Section 3.1 and provide further recommendations in Appendix B. We then provide motivation for several of our modeling choices in Section 3.2.

### 3.1 Practical Considerations

**Working with transformations.** Repeated application of transformations—e.g., in Figure 4—can introduce unwanted artifacts such as blurring. For many useful transformations, we can compose transformations before applying them. For affine transformations of images, for example, we can directly multiply affine-transformation matrices. More generally, if there is some representation of the transformation parameters $T(\boldsymbol{\eta})$ where composition can be performed—e.g., as matrix multiplication $\mathcal{T}_{\boldsymbol{\eta}_2} \circ \mathcal{T}_{\boldsymbol{\eta}_1} = \mathcal{T}'_{T(\boldsymbol{\eta}_2)T(\boldsymbol{\eta}_1)}$, in the case where $T$ is a group representation—then we recommend composing transformations in that space to minimize the number of applications.

**Partial invertibility.** In many common settings, transformations are not fully invertible. We encounter two such issues when working with affine transformations of images living in a finite, discrete coordinate space. Firstly, affine transformations are only *approximately* invertible in the discrete space due to the information loss when interpolating the transformed image onto a discrete grid. Thus, while only a single prototype $\hat{x}$ exists for any $\mathbf{x}$, it may not be clear what the correct prototype is. Secondly, transformations can cause information loss due to the finite coordinate space (e.g., by shifting the contents of the image out-of-bounds[4]). If appropriate bounds are known *a priori*, we can prevent severe information loss by constraining $\boldsymbol{\eta}_{\text{min}}$ and $\boldsymbol{\eta}_{\text{max}}$ using tanh, scale, and shift bijectors. Alternatively, we can augment the SSL loss in Algorithm 1 with an *invertibility loss*

$$\mathcal{L}_{\text{invertibility}}(\boldsymbol{\omega}) = \texttt{mse}\left(\boldsymbol{x}, \mathcal{T}^{-1}_{f_{\boldsymbol{\omega}}(\mathbf{x})}\left(\mathcal{T}_{f_{\boldsymbol{\omega}}(\mathbf{x})}(\boldsymbol{x})\right)\right). \tag{7}$$

**Learning $p_\psi(\eta \mid \hat{x})$ with imperfect inference.** In practice, our transformation inference network $f_{\boldsymbol{\omega}}(\mathbf{x})$ will not be perfect; see Figure 10. Even after training, there may be small variations in the prototypes $\hat{x}$ corresponding to different elements in the orbit of $\mathbf{x}$. To make $p_\psi(\boldsymbol{\eta}_{\mathbf{x}} \mid \hat{x})$ robust to these variations, we train it with prototypes corresponding to *randomly transformed* training data points. I.e., we modify the MLE objective in Algorithm 1 as $\log p_\psi(\boldsymbol{\eta}_{\boldsymbol{x}} \mid \hat{\boldsymbol{x}}')$, where $\hat{\boldsymbol{x}}' = \mathcal{T}^{-1}_{f_{\boldsymbol{\omega}}(\mathcal{T}_{\boldsymbol{\eta}_{\text{rnd}}}(\boldsymbol{x}))}(\mathcal{T}_{\boldsymbol{\eta}_{\text{rnd}}}(\boldsymbol{x}))$ as in our SSL objective. Averaging the loss over multiple samples—e.g., 5—of $\boldsymbol{\eta}_{\text{rnd}}$ is beneficial.

### 3.2 Modelling Choices

We now motivate some of the design choices for our SGM by means of illustrative examples. In each case, we assume that $\mathcal{T}_\eta$ is counter-clockwise rotation; thus, $\eta$ is the angle.

**1. The distribution $p_\psi(\eta \mid \hat{x})$ is implemented as a normalising flow.** Consider a dataset of '8's rotated in the range $-30°$ to $30°$: $\{ ၆, \ldots, 8, \ldots, 8 \}$. Let us assume that the prototype is '8'. Figure 5a shows $p(\eta \mid \mathbf{x}, \hat{x})$, an example of the true distribution for $\eta$ given $\mathbf{x}$ and $\hat{x}$, for several observations, under the data generating process[5]. These distributions are composed of deltas because

---

[4]This can occur in practice since our SSL objective—which aims to make prototypes as similar as possible—can trivially be minimized by removing all of the contents of an image.

[5]Because '8' is symmetric, $p(\eta \mid \mathbf{x}, \hat{x})$ could be any convex combination of the two delta distributions. However, for a more realistic example, consider a prototype '8' with a smaller upper loop. In this case, the $p(\eta \mid \hat{x})$ must be bimodal to capture '8's with both smaller upper and lower loops.

(a) Distribution for η given **x** and $\hat{x}$.

| $x$ | $\hat{x}$ | $p(\eta \mid \boldsymbol{x}, \hat{\boldsymbol{x}})$ |
|---|---|---|
| 2 | 2 | $\delta(\eta - 0°)$ |
| ↶2 | 2 | $\delta(\eta - 30°)$ |
| ↷2 | 2 | $\delta(\eta + 30°)$ |
| 8 | 8 | $0.5 \cdot \delta(\eta - 0°) + 0.5 \cdot \delta(\eta - 180°)$ |
| ↶8 | 8 | $0.5 \cdot \delta(\eta - 30°) + 0.5 \cdot \delta(\eta + 150°)$ |
| ↷8 | 8 | $0.5 \cdot \delta(\eta + 30°) + 0.5 \cdot \delta(\eta - 150°)$ |

(b) η          (c) η | $\hat{x}$

Figure 6: Examples of learned distributions over angles $p_\psi(\cdot)$— ▨ —with and without dependence on $\hat{x}$, given the true distribution $p(\cdot)$— ↑↑↑↑↑↑ .

only certain values of η will transform $\hat{x}$ into **x**. Figures 5b and 5c compare idealised examples of the learned $p_\psi(\eta \mid \hat{x})$—given a *simple* uni-modal Gaussian family and a more *flexible* bi-modal mixture-of-Gaussian family—with the aggregate true distribution $p(\eta \mid \hat{x}) = \sum_{\mathbf{x} \in \{↶8,...,8,...,↷8\}} p(\eta \mid \mathbf{x}, \hat{x})$. Here, the simple uni-modal distribution is clearly worse than the bi-modal distribution due to the large amount of probability mass being wasted on angles with low density under the true data-generating process. Of course, one might argue that the bi-modal distribution is also not flexible enough. Furthermore, 'flexible enough' will be problem-specific. We solve this problem with normalizing flows, which can match a wide range of distributions.

**2. The transformation parameters η depend on the prototype $\hat{x}$.** Consider a dataset of '2's and '8's rotated in the range $-30°$ to $30°$: $\{↶2, ..., 2, ..., ↷2, ↶8, ..., 8, ..., ↷8\}$, with prototypes '2' and '8'. Figure 6a shows $p(\eta \mid \mathbf{x}, \hat{x})$, an example of a true distribution over η, for several observations. Figures 6b and 6c compares idealised examples of learned distributions over η and η | $\hat{x}$. Without dependence on $\hat{x}$, the model must place probability mass between $-150°$ and $150°$, in order to capture the symmetries of the '8's, however this results invalid digits—such as $\{↶c, ↷c, ↶c\}$—which do not come from true data distribution. On the other hand, when η depends on $\hat{x}$, the distribution conditioned on the prototype for the '2's only needs to place mass in $[-30°, 30°]$.

**3. The prototype $\hat{x}$ is *fully* invariant to transformations of x.** Models such as CNNs are most useful when we know *a priori* which symmetries are present in the data. However, in many cases, this must be learned. In the case of handwritten digit recognition, we know that the model should be invariant to some amount of rotation since people naturally write with some variation in angle. But a model that is invariant to rotations in the full range $[-180°, 180°]$ might be unable to distinguish between '6' and '9'. Thus, in the literature for learning invariances in the discriminative setting, it is common to learn *partially* invariant functions that capture some degree of invariance [van der Wilk et al., 2018, Benton et al., 2020, van der Ouderaa and van der Wilk, 2022]. However, as we will now show, this approach is unsuitable for our SGM, as it breaks our assumption that $\hat{x}$ contains no information about the symmetries in the data.

(a) $p(\eta \mid \mathbf{x}, \hat{x})$ with different levels of invariance.

| | | (a) FULL | | (b) PARTIAL | | (c) NONE |
|---|---|---|---|---|---|---|
| $x$ | $\hat{x}$ | $p(\eta \mid \boldsymbol{x}, \hat{\boldsymbol{x}})$ | $\hat{x}$ | $p(\eta \mid \boldsymbol{x}, \hat{\boldsymbol{x}})$ | $\hat{x}$ | $p(\eta \mid \boldsymbol{x}, \hat{\boldsymbol{x}})$ |
| 2 | 2 | $\delta(\eta - 0°)$ | ↶2 | $\delta(\eta + 15°)$ | 2 | $\delta(\eta - 0°)$ |
| ↶2 | 2 | $\delta(\eta - 30°)$ | ↶2 | $\delta(\eta - 15°)$ | ↶2 | $\delta(\eta - 0°)$ |
| ↷2 | 2 | $\delta(\eta + 30°)$ | ↷2 | $\delta(\eta - 0°)$ | ↷2 | $\delta(\eta - 0°)$ |

(b) FULL          (c) PARTIAL          (d) NONE

Figure 7: Examples of learned distributions over angles $p_\psi(\eta \mid \hat{x})$— ▨ / ↑ —with different degrees of invariance in the prototype $\hat{x}$, given the true $p(\eta \mid \hat{x})$— ↑↑↑ .

Consider a dataset of '2's rotated in the range $-30°$ to $30°$: $\{↶2, ..., 2, ..., ↷2\}$. Figure 7a shows predicted prototypes and the corresponding distributions over η for several observations. There are three cases: **(a)** a fully-invariant $\hat{x}$, i.e., there is a single prototype, **(b)** a partially-invariant $\hat{x}$, for which there are two prototypes in this example, and **(c)** a non-invariant $\hat{x}$, which takes the partially-invariant case to the extreme and has as many prototypes as observations. In the partially-invariant and non-invariant cases, we can get multiple prototypes rather than a single unique prototype per orbit, which is invalid under the generative model of the data. As a result, $p_\psi(\eta \mid \hat{x})$ does not represent the distribution of naturally occurring transformations of $\hat{x}$ in the data. This is illustrated in Figures 7b to 7d, which show idealized examples of the learned $p_\psi(\eta \mid \hat{x})$ in each case. While the distribution in Figure 7b matches the distribution of transformations in the dataset, in Figures 7c and 7d we see that the distributions corresponding to non-unique prototype do not. To illustrate why this is a problem, let us say we would like to probe the probability of a particular transformed variant

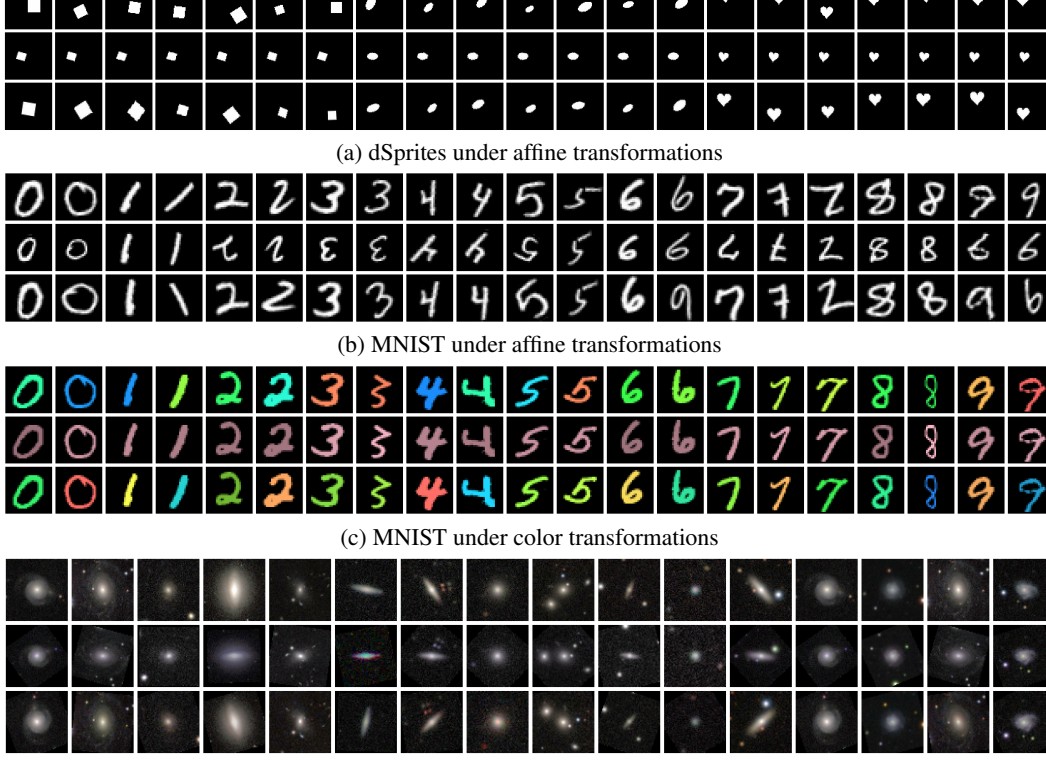

(a) dSprites under affine transformations

(b) MNIST under affine transformations

(c) MNIST under color transformations

(d) GalaxyMNIST under affine and color transformations

Figure 8: **Top:** samples from the test set. **Mid:** prototypes for each test example. **Bot:** resampled versions of each test example given the prototype. Prototypes for examples from the same orbit (and in some cases from distinct but similar orbits) match (e.g., their size, position, rotation, etc. are similar). Resampled examples are usually indistinguishable from test examples.

of an observed example. For example, given an example ▨ of a digit '3', we want to know the probability of observing ⟳, that digit rotated by -90°. Assuming we can find a prototype $\hat{x}$ we would like $p(\eta \,|\, \hat{\mathbf{x}} = \hat{x})$ to represent all naturally occurring augmentations. Unless $\hat{x}$ is unique, this won't necessarily be the case, as illustrated in Figure 7.

## 4 Experiments

In Section 4.1, we explore our SGM's ability to learn symmetries. We show that it produces valid prototypes, and generates plausible samples from the data distribution, given those prototypes. Then, in Section 4.2, we leverage our SGM to improve data efficiency in deep generative models.

We conduct experiments using three datasets—dSprites [Matthey et al., 2017], MNIST, and GalaxyMNIST [Walmsley et al., 2022]—and two kinds of transformations—affine and color. In Section 4.1, when working with MNIST under affine transformations, we add a small amount of rotation (in the range $[-15°, 15°]$) to the original data to make rotations in the figures easier to see. For MNIST under color transformations, we first convert the grey-scale images to color images using only the red channel. We then add a random hue rotation in the range $[0, 0.6\pi]$ and a random saturation multiplier in the range $[0.6, 0.9]$. In the case of dSprites, we carefully control the rotations, positions, and sizes of all of the sprites. For example, in the case of the heart sprites, we have removed the rotations and set the $y$-positions to be bimodal in the top and bottom of the images. Further details about the dSprites setup, as well as all other experimental details, can be found in Appendix C. We focus on learning affine transformations (shifting, rotation, and scaling) as they are expressive while still being a group that is easy to work with. We also learn color transformations (hue, saturation, and value). See Appendix C.7 for details about how we parameterize $\mathcal{T}_\eta$ in both cases.

### 4.1 Learning Symmetries

**Exploring transformations and prototypes.** Figure 8 shows that for both datasets and kinds of transformations we consider, our SGM produces close-to-invariant prototypes as well as realistic "natural" examples that are almost indistinguishable from test examples. There are sev-

eral illustrative examples which bear further discussion. The heart sprites in Figure 8a show that our SGM was able to learn *the absence* of a transformation (namely rotation) in the dataset.

As expected, all of the prototypes for the sprites of the same shape are the same, since these shapes are in the same orbit as one another. This behaviour is also demonstrated for MNIST digits in Figures 19 and 20. The '6', '8', and '9' digits in Figure 8b demonstrate the ability of our SGM to learn bimodal distributions (on rotation in this case). The figure's third '7' is interesting because our SGM interprets it as a '2'.

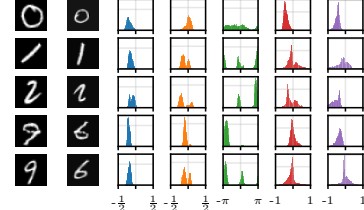

Figure 9: From left to right, test examples, their prototypes, and the corresponding marginal distributions $p_\psi(\eta_i \mid \mathbf{x})$ over translation in $x$, translation in $y$, rotation, scaling in $x$, and scaling in $y$.

**Flexibility is important.** In $\eta$, each dimension corresponds to a different transformation. We refer to $p_\psi(\eta_i \mid \mathbf{x})$ as the marginal distribution of a single transformation parameter. Figure 9 shows these marginal learnt distributions for several digits from Figure 8b. We see that each of the parameters has its own range and shapes. For rotations, which are easy to reason about, we see distributions that make sense—the round '0' has an almost uniform distribution over rotations, and the '1' and one of the '9's are strongly bimodal as expected. The other '9', which does not look as much like an upside-down '6', has a much smaller 2nd mode. The '2', which looks somewhat like an upside-down '7', is also bimodal. We see that prototypes of different sizes result in corresponding distributions over scaling parameters with different ranges. Figure 21 provides additional examples for MNIST with affine transformations, while Figure 22 provides the same for color transformations, and Figure 23 investigates the distributions for dSprites. These results provide experimental evidence of the need for flexibility in the generative model for $p_\psi(\eta \mid \mathbf{x})$, as conjectured in Section 3.2. We also find significant dependencies between dimensions of $\eta$ (e.g., rotation and translation in dSprites).

**Invariance of $f_\omega$ and the prototypes.** In Figure 10, we investigate the imperfections of the inference network by considering an iterative procedure in which prototypes are treated as observed examples, allowing us to infer a chain of successive prototypes. We show several examples of such chains, as well as the average magnitude of the transformation parameters at each iteration, normalized by the maximum magnitude (at iteration 0). The first prototype $\hat{\mathbf{x}}_1$ is most different from the previous $\hat{\mathbf{x}}_0 = \mathbf{x}$, with successive prototypes being similar visually and as measured by the magnitude of the inferred transformation parameters. However, the magnitude of the inferred parameters does not tend towards 0, rather plateauing at around 5% of the maximum. This highlights that, although simple NNs can learn to be approximately invariant, a natively invariant architecture has the potential to improve performance.

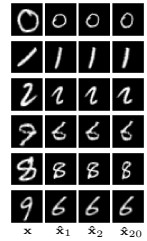 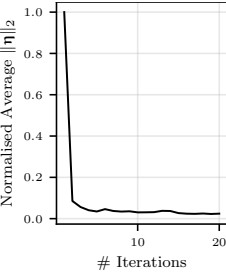

Figure 10: Iterative prototype inference. **Left:** starting with a test example $\mathbf{x}$, we get a prototype $\hat{\mathbf{x}}_1$, then treating prototype $\hat{\mathbf{x}}_i$ as an observed example we predict the next prototype $\hat{\mathbf{x}}_{i+1}$. **Right:** The average magnitude of the transformation parameters as a function of iterations of this process.

## 4.2 VAE Data Efficiency

We use SGM to build data-efficient and robust generative models. In Figure 11, we compare a standard VAE to two VAE-SGM hybrid models—"AugVAE" and "InvVAE"—for different amounts of training data and added rotation of the MNIST digits. When adding rotation, each $x$ in the dataset set is always rotated by the same angle (sampled uniformly between $\pm\theta_{\max}$, the maximum added rotation angle). Thus, adding rotation here is *not* data augmentation. AugVAE is a VAE that uses our SGM to re-sample transformed examples $x' = \mathcal{T}_{\eta\mid\hat{x}}(\hat{x})$, introducing data augmentation at training time. InvVAE is a VAE that uses our SGM to convert each example $x$ to its prototype $\hat{x}$ at both train and test time. That is, the VAE in InvVAE sees only the invariant representation of each example. We also compare against a VAE trained with standard data augmentation[6]. We use test-set importance-weighted lower bound (IWLB) [Domke and Sheldon, 2018] of $p(\mathbf{x})$, estimated with 300 samples of the VAE's latent variable $\mathbf{z}$, and $\eta$ for InvVAE, to compare the models. Reconstruction error is provided in Appendix E. Further details—e.g., hyperparameter sweeps—are in Appendix C.

---

[6]We use rotation $\sim \mathcal{U}(-15°, 15°)$, zoom $\sim \mathcal{U}(-10\%, 10\%)$, and x/y-shift $\sim \mathcal{U}(-2\text{px}, 2\text{px})$.

As expected, for the VAE ( ┈┈ ), as we decrease the amount of training data ( ━ → ⋯ ) or increase the amount of randomly added rotation, performance degrades. This is because the VAE sees fewer training examples *per-degree of rotation*. On the other hand, the AugVAE ( ┈┈ ) is more data efficient. Its performance is unaffected by reducing the number of observations by three quarters. Furthermore, while the performance of AugVAE and the standard VAE are almost identical for small angles and large training sets, the drop in performance of AugVAE for larger random rotations is significantly smaller; AugVAE *does not* see less training examples *per-degree of rotation*. InvVAE ( ┈┈ ), which natively incorporates the inductive biases of our SGM and obtains a 500 nat larger likelihood than the other models. Its performance is almost perfectly robust to rotation in the dataset. Additionally, its metrics barely change ($< 10\%$) when trained on half the data. Finally, while the VAE with data augmentation ( ┈┈ ) improves on the standard VAE for less training data, it is substantially worse in the presence of more data. This contrasts our AugVAEs, which are almost always better. This poor performance is because the augmentations are independent of the samples. Thus, highly rotated digits can be rotated

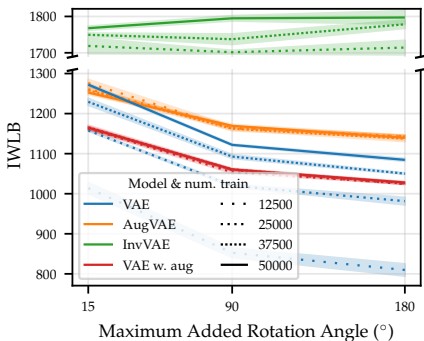

Figure 11: **Incorporating symmetries improves data efficiency.** Importance-weighted lower bound (IWLB) (mean and std. err. over 3 random seeds) on rotated MNIST for a standard VAE (w. and w.o. data aug.) and two VAE variants that incorporate symmetries via our SGM. Improved data efficiency is demonstrated by better performance with less training data and less sensitivity to added rotation.

too much, smaller digits become too small, and digits near the image edges are moved out of frame. This highlights the importance of augmenting data in accordance with the true data distribution.

We further validate these results with the more complex GalaxyMNIST dataset and an enlarged set of both affine and color transformations. As with our rotated MNIST with affine transformation results, in Figure 12, we see that AugVAE ( ▬ ) outperforms the standard VAE ( ▬ ). Furthermore, we see that AugVAE is robust to training with only half of the dataset. Our SGM captures the true data distribution with only 3500 training examples.

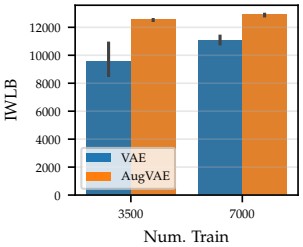

Figure 12: GalaxyMNIST data-efficiency (3 seed mean & std. err.).

## 5   Related Work

**Learning Lie groups.**   Rao and Ruderman [1998], Miao and Rao [2007], Keurti et al. [2023] learn Lie groups from sequences of transformed images in an unsupervised fashion. Hashimoto et al. [2017] learn to represent an image as a linear combination of transformed versions of its nearest neighbors. Dehmamy et al. [2021] use Lie algebras to define CNNs for automatic symmetry discovery. Yang et al. [2023] use a GAN-based approach to learn transformations of examples that leave the original data distribution unchanged, thereby fooling a discriminator. Falorsi et al. [2019] introduce a reparameterization trick for learning densities on arbitrary, but known, Lie groups. Chau et al. [2022] learn a generative model over Lie group transformations applied to prototypical images that are themselves composed of sparse combinations of learned dictionary elements.

**Learning a prototype.**   Kaba et al. [2023] note that symmetry-based NNs are often contained in their architectures. Like us, they propose to learn "canonicalization functions" that produce prototypical representations of the data. Mondal et al. [2023] show that such canonicalization functions can be used to make large-pre-trained NNs equivariant and, when combined with dataset-dependent symmetry priors, do not degrade performance. Similarly, Kim et al. [2023] learn architecture-agnostic equivariant functions by averaging a non-equivariant function over a probabilistic prototypical input. Finally, while not explicitly trained to produce prototypes, spatial transformers learn to undo transformations such as translation, scaling, and rotations [Jaderberg et al., 2015].

**Data augmentations and symmetries.**   Prior work makes several connections between data augmentation and symmetries relevant to our findings. Bouchacourt et al. [2021b] show that invariances in the model tend to result from natural variations in the data rather than data augmentation or model architecture. This supports our approach of learning data augmentation from the data and our

architecture-agnostic self-supervised invariance learning method. Balestriero et al. [2022], Miao et al. [2023], Bouchacourt et al. [2021b] show that learned symmetries (i.e., data augmentation) should be class-dependent, much like our transformations are prototype-dependent.

**Symmetry-aware latent spaces.** Encoding symmetries in latent space is well-studied. Higgins et al. [2018] posit that symmetry transformations that leave some parts of the world invariant are responsible for exploitable structure in any dataset. Thus, agents benefit from *disentangled* representations that separate out these transformations. Winter et al. [2022] split the latent space of an auto-encoder into invariant and equivariant partitions. However, they rely on geometric NN architectures, contrasting with our self-supervised learning approach. Furthermore, they do not learn a generative model—they reconstruct the input exactly—thus, they cannot sample new observations given a prototype. Xu et al. [2021] propose group equivariant subsampling layers that allow them to construct autoencoders with equivariant representations. Shu et al. [2018] propose an autoencoder whose representations are split such that the reconstruction of an observation is decomposed into a "template" (much like our prototypes) and a spatial deformation (transformation).

In the generative setting, Louizos et al. [2016] construct a VAE with a latent space that is invariant to pre-specified sensitive attributes of the data. However, these sensitive attributes are observed rather than learned. Similarly, Aliee et al. [2023] construct a VAE with a partitioned latent space with a component that is invariance spurious factors of variation in the data. Bouchacourt et al. [2018], Hosoya [2019] learn VAE with two latent spaces—a per-observation equivariant latent and an invariant latent shared across grouped examples. Other works have constructed rotation equivariant [Kuzina et al., 2022] and partitioned equivariant and invariant [Vadgama et al., 2022] latent spaces. Antorán and Miguel [2019], Ilse et al. [2020] split the latent space of a VAE into domain, class, and residual variation components. The first of which can capture rotation symmetry in hand-written digits. Unlike us, they require class labels and auxiliary classifiers. Keller and Welling [2021] construct a VAE with a topographically organised latent space such that an approximate equivariance is learned from sequences of observations. In contrast to the works above, Bouchacourt et al. [2021a] argue that learning symmetries should not be achieved via a partitioned latent space but rather learning *equivariant operators* that are applied to the whole latent space. Finally, while Nalisnick and Smyth [2017] do not learn symmetries, their *information lower bound* objective is reminiscent of several works above—and our own, see Appendix A—in minimizing the mutual information between two quantities when learning a prior.

**Self-supervised Equivariant Learning** [Dangovski et al., 2022] generalize standard invariant SSL methods to produce representations that can be either insensitive (invariant) or sensitive (equivariant) to transformations in the data. Similarly, Eastwood et al. [2023] use a self-supervised learning approach to disentangle sources of variation in a dataset, thereby learning a representation that is equivariant to each of the sources while invariant to all others.

# 6 Conclusion

We have presented a Symmetry-aware Generative Model (SGM) and demonstrated that it is able to learn, in an unsupervised manner, a distribution over symmetries present in a dataset. This is done by modeling the observations as a random transformation of an invariant latent *prototype*. This is the first such model we are aware of. Building generative models that incorporate this understanding of symmetries significantly improves log-likelihoods and data sparsity robustness. This is exciting in the context of modern generative models, which are close to exhausting all of the data on the internet. We are also excited about the use of SGM for scientific discovery, given that the framework is ideal for probing for naturally occurring symmetries present in systems. For example, we could apply SGM to marginalize out the idiosyncrasies of different measuring equipment and observation geometry in radio astronomy data. Additionally, given the success of using our SGM for data augmentation when training VAEs, it would be interesting to apply it to data augmentation in discriminative settings and compare it with methods such as Benton et al. [2020], Miao et al. [2023].

The main limitation of our SGM is that it requires specifying the super-set of possible symmetries. Future work might relax this requirement or explore how robust our SGM is to even larger sets. Furthermore, care must sometimes be taken when specifying the set of symmetries. For example, when rotating to images with "content" up to the boundaries of the image; see Appendix E.2.

## Acknowledgements

The authors would like to thank Taliesin Beynon for helpful discussions and Emile Mathieu for providing feedback on the paper. This work has been performed using resources provided by the Cambridge Tier-2 system operated by the University of Cambridge Research Computing Service (http://www.hpc.cam.ac.uk) funded by EPSRC Tier-2 capital grant EP/T022159/1. This work was also supported with Cloud TPUs from Google's TPU Research Cloud (TRC). JUA acknowledges funding from the EPSRC, the Michael E. Fisher Studentship in Machine Learning, and the Qualcomm Innovation Fellowship. JUA was also supported by an ELLIS mobility grant. SP acknowledges support from the Harding Distinguished Postgraduate Scholars Programme Leverage Scheme. JA acknowledges support from Microsoft Research, through its PhD Scholarship Programme, and from the EPSRC. JMH acknowledges support from a Turing AI Fellowship under grant EP/V023756/1. RET is supported by Google, Amazon, ARM, Improbable, EPSRC grant EP/T005386/1, and the EPSRC Probabilistic AI Hub (ProbAI, EP/Y028783/1).

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

# A Connections to MLL Optimization

As we will now show, Algorithm 1 has connections to marginal log-likelihood (MLL) maximization via VAE-like amortized inference. Given the graphical model in Figure 2, we can derive an Evidence Lower BOund (ELBO) for jointly learning the generative and inference parameters with gradients:

$$\log p(\mathbf{x}) = \log \iint p(\mathbf{x}, \boldsymbol{\eta}, \hat{\mathbf{x}}) \, d\boldsymbol{\eta} \, d\hat{\mathbf{x}} \tag{8}$$

$$= \log \iint p(\mathbf{x} \,|\, \boldsymbol{\eta}, \hat{\mathbf{x}}) \, p_{\boldsymbol{\psi}}(\boldsymbol{\eta} \,|\, \hat{\mathbf{x}}) \, p_{\boldsymbol{\theta}}(\hat{\mathbf{x}}) \, d\boldsymbol{\eta} \, d\hat{\mathbf{x}}$$

$$= \log \iint p(\mathbf{x} \,|\, \boldsymbol{\eta}, \hat{\mathbf{x}}) \, p_{\boldsymbol{\psi}}(\boldsymbol{\eta} \,|\, \hat{\mathbf{x}}) \, p_{\boldsymbol{\theta}}(\hat{\mathbf{x}}) \frac{q_{\boldsymbol{\omega}}(\boldsymbol{\eta}, \hat{\mathbf{x}} \,|\, \mathbf{x})}{q_{\boldsymbol{\omega}}(\boldsymbol{\eta}, \hat{\mathbf{x}} \,|\, \mathbf{x})} d\boldsymbol{\eta} \, d\hat{\mathbf{x}} \tag{9}$$

$$= \log \mathop{\mathbb{E}}_{q_{\boldsymbol{\omega}}(\boldsymbol{\eta}, \hat{\mathbf{x}} \,|\, \mathbf{x})} \left[ \frac{p(\mathbf{x} \,|\, \hat{\mathbf{x}}, \boldsymbol{\eta}) \, p_{\boldsymbol{\psi}}(\boldsymbol{\eta} \,|\, \hat{\mathbf{x}}) \, p_{\boldsymbol{\theta}}(\hat{\mathbf{x}})}{q_{\boldsymbol{\omega}}(\boldsymbol{\eta}, \hat{\mathbf{x}} \,|\, \mathbf{x})} \right] \tag{10}$$

$$\geq \underbrace{\mathop{\mathbb{E}}_{q_{\boldsymbol{\omega}}(\boldsymbol{\eta}, \hat{\mathbf{x}} \,|\, \mathbf{x})} [\log p(\mathbf{x} \,|\, \boldsymbol{\eta}, \hat{\mathbf{x}})]}_{\text{likelihood}} - \underbrace{D_{\mathrm{KL}} [q_{\boldsymbol{\omega}}(\boldsymbol{\eta}, \hat{\mathbf{x}} \,|\, \mathbf{x}) \,||\, p_{\boldsymbol{\psi}}(\boldsymbol{\eta} \,|\, \hat{\mathbf{x}}) \, p_{\boldsymbol{\theta}}(\hat{\mathbf{x}})]}_{\text{KL-divergence}} \tag{11}$$

$$\equiv -\mathcal{L}(\boldsymbol{\theta}, \boldsymbol{\psi}, \boldsymbol{\omega}), \tag{12}$$

where $p_{\boldsymbol{\theta}}(\hat{\mathbf{x}})$ is some generative model—e.g., a VAE—for prototypes, with parameters $\boldsymbol{\theta}$, and $q_{\boldsymbol{\omega}}(\boldsymbol{\eta}, \hat{\mathbf{x}} \,|\, \mathbf{x}) = q_{\boldsymbol{\omega}}(\boldsymbol{\eta} \,|\, \mathbf{x}) \, p(\hat{\mathbf{x}} \,|\, \mathbf{x}, \boldsymbol{\eta})$. Now, we can show that the gradient of the *likelihood* term in the ELBO is approximated by the gradient of our SSL loss on line 1 of Algorithm 1:

$$\nabla_{\boldsymbol{\omega}} \mathop{\mathbb{E}}_{q_{\boldsymbol{\omega}}(\boldsymbol{\eta} \,|\, \mathbf{x}) p(\hat{\mathbf{x}} \,|\, \mathbf{x}, \boldsymbol{\eta})} [\log p(\mathbf{x} \,|\, \hat{\mathbf{x}}, \boldsymbol{\eta})] \tag{13}$$

$\triangleright p(\mathbf{x} \,|\, \hat{\mathbf{x}}, \boldsymbol{\eta}) = \delta(\mathbf{x} - \mathcal{T}_{\boldsymbol{\eta}}(\hat{\mathbf{x}})) = \lim_{\sigma^2 \to 0} \mathcal{N}(\mathbf{x} \,|\, \mathcal{T}_{\boldsymbol{\eta}}(\hat{\mathbf{x}}), \sigma^2)$:

$$\approx \nabla_{\boldsymbol{\omega}} \mathop{\mathbb{E}}_{q_{\boldsymbol{\omega}}(\boldsymbol{\eta} \,|\, \mathbf{x}) p(\hat{\mathbf{x}} \,|\, \mathbf{x}, \boldsymbol{\eta})} \left[ \log \mathcal{N}(\mathbf{x} \,|\, \mathcal{T}_{\boldsymbol{\eta}}(\hat{\mathbf{x}}), \sigma^2) \right] \tag{14}$$

$\triangleright$ take 1 sample, $\boldsymbol{\eta} \sim q_{\boldsymbol{\omega}}(\boldsymbol{\eta} \,|\, \boldsymbol{x})$:

$$\approx \nabla_{\boldsymbol{\omega}} \log \mathcal{N}(\boldsymbol{x} \,|\, \mathcal{T}_{\boldsymbol{\eta}}(\hat{\boldsymbol{x}}), \sigma^2), \tag{15}$$

$\triangleright$ definition of Gaussian PDF:

$$= \nabla_{\boldsymbol{\omega}} - 0.5 \, \|\boldsymbol{x} - \mathcal{T}_{\boldsymbol{\eta}}(\hat{\boldsymbol{x}})\|_2^2 / \sigma^2 - \log\left(\sqrt{2\pi}\sigma\right) \tag{16}$$

$\triangleright$ drop constant term:

$$= \nabla_{\boldsymbol{\omega}} - 0.5 \, \texttt{mse}(\boldsymbol{x}, \mathcal{T}_{\boldsymbol{\eta}}(\hat{\boldsymbol{x}})) / \sigma^2. \tag{17}$$

The negative sign is due to the fact that the ELBO is maximized, whereas our SSL loss is minimized. The gradient of the *KL-divergence* term w.r.t. $\boldsymbol{\psi}$ is approximated by the gradient of our MLE loss on line 8 of Algorithm 1:

$$\nabla_{\boldsymbol{\psi}} D_{\mathrm{KL}} [q_{\boldsymbol{\omega}}(\boldsymbol{\eta}, \hat{\mathbf{x}} \,|\, \mathbf{x}) \,||\, p_{\boldsymbol{\psi}}(\boldsymbol{\eta} \,|\, \hat{\mathbf{x}}) \, p_{\boldsymbol{\theta}}(\hat{\mathbf{x}})] \tag{18}$$

$\triangleright$ definition of $D_{\mathrm{KL}}$:

$$= \nabla_{\boldsymbol{\psi}} \mathop{\mathbb{E}}_{q_{\boldsymbol{\omega}}(\boldsymbol{\eta} \,|\, \mathbf{x}) p(\hat{\mathbf{x}} \,|\, \mathbf{x}, \boldsymbol{\eta})} \left[ \log \frac{q_{\boldsymbol{\omega}}(\boldsymbol{\eta} \,|\, \mathbf{x}) \, p(\hat{\mathbf{x}} \,|\, \mathbf{x}, \boldsymbol{\eta})}{p_{\boldsymbol{\psi}}(\boldsymbol{\eta} \,|\, \hat{\mathbf{x}}) \, p_{\boldsymbol{\theta}}(\hat{\mathbf{x}})} \right] \tag{19}$$

$\triangleright$ drop constant terms and use $\hat{\mathbf{x}} = \mathcal{T}_{\boldsymbol{\eta}}^{-1}(\mathbf{x})$ :

$$= \nabla_{\boldsymbol{\psi}} \mathop{\mathbb{E}}_{q_{\boldsymbol{\omega}}(\boldsymbol{\eta} \,|\, \mathbf{x})} \left[ -\log p_{\boldsymbol{\psi}}\left( \boldsymbol{\eta} \,\middle|\, \mathcal{T}_{\boldsymbol{\eta}}^{-1}(\mathbf{x}) \right) \right] \tag{20}$$

$\triangleright$ take 1 sample, $\boldsymbol{\eta}_x \sim q_{\boldsymbol{\omega}}(\boldsymbol{\eta} \,|\, \mathbf{x})$:

$$\approx \nabla_{\boldsymbol{\psi}} - \log p_{\boldsymbol{\psi}}\left( \boldsymbol{\eta}_x \,\middle|\, \mathcal{T}_{\boldsymbol{\eta}_x}^{-1}(\boldsymbol{x}) \right). \tag{21}$$

Note that the sampling approximations in both (15) and (21) also apply to VAE-like amortized inference algorithms.

While ELBO training and our algorithm share some similarities, some key differences exist. For instance, we do not learn the generative and inference models jointly. This disjoint training is equivalent

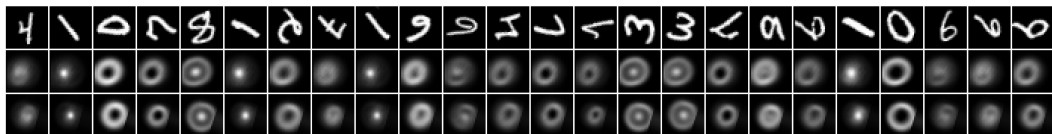

Figure 13: Failure of an invariant VAE encoder. **Top:** MNIST digits sampled from the test set. **Mid:** Prototypes produced by VAE who's encoder is made invariant using (22), where $\boldsymbol{\eta} \sim \mathcal{U}\left(-\boldsymbol{\eta}_{\max}, \boldsymbol{\eta}_{\max}\right)$ and $\boldsymbol{\eta}_{\max} = (0.25, 0.25, \pi, 0.25, 0.25)$. **Bot:** Reconstructed digits. The model becomes stuck in a local optima where the prototypes and 'reconstructions' are all circles and rings of various sizes depending on the input image. The averaged latent code is free of (e.g.,) rotation information but has also lost almost all information that identifies each digit.

to ignoring the gradient $\nabla_{\boldsymbol{\omega}} D_{\mathrm{KL}}\left[q_{\boldsymbol{\omega}}\left(\boldsymbol{\eta}, \hat{\mathbf{x}} \mid \mathbf{x}\right) \| p_{\boldsymbol{\psi}}\left(\boldsymbol{\eta} \mid \hat{\mathbf{x}}\right) p_{\boldsymbol{\theta}}\left(\hat{\mathbf{x}}\right)\right]$ when training $q_{\boldsymbol{\omega}}\left(\boldsymbol{\eta} \mid \mathbf{x}\right)$. This KL-divergence has two components: entropy $-\mathbb{H}\left[q_{\boldsymbol{\omega}}\right]$ and cross entropy $\mathbb{H}\left[q_{\boldsymbol{\omega}}, p_{\boldsymbol{\psi}} p_{\boldsymbol{\theta}}\right]$. Assuming that $p_{\boldsymbol{\psi}}\left(\boldsymbol{\eta} \mid \hat{\mathbf{x}}\right)$ is sufficiently flexible, the cross entropy term should not have a significant impact on $q_{\boldsymbol{\omega}}\left(\boldsymbol{\eta} \mid \mathbf{x}\right)$ since $p_{\boldsymbol{\psi}}$ is trained to match $q_{\boldsymbol{\omega}}$. On the other hand, $q_{\boldsymbol{\omega}}\left(\boldsymbol{\eta} \mid \mathbf{x}\right)$ should be close to a delta since there should be a single prototype for each $\mathbf{x}$. Thus, encouraging high variance with an entropy term might actually be harmful. Another difference is that we do not need to learn $p_{\boldsymbol{\theta}}\left(\hat{\mathbf{x}}\right)$, which has the benefit that we can learn the symmetries in a dataset without having to learn to generate the data itself, greatly simplifying training for the complicated dataset. Furthermore, actually evaluating the gradient of the likelihood term in (12) is challenging due to the fact that $p\left(\mathbf{x} \mid \hat{\mathbf{x}}, \boldsymbol{\eta}\right)$ is a delta.

Given all of these differences, it might be natural to question the utility of the comparison between our algorithm and maximization of (12). Perhaps the most useful connection to draw is that of Equations (18) and (21), which motivates our MLE learning objective for $p_{\boldsymbol{\omega}}\left(\boldsymbol{\eta} \mid \hat{\mathbf{x}}\right)$ as being closely related to the process of learning a prior in an ELBO.

In an early version of this work [Allingham et al., 2022], we trained a variant of the SGM using an ELBO similar to (12), with the main difference being that $\hat{\mathbf{x}}$ was modeled using a VAE and invariance was incorporated into the VAE encoder. We constructed an invariant encoder $q_{\boldsymbol{\phi}}\left(\mathbf{z} \mid \mathbf{x}\right)$ from a non-invariant encoder $\hat{q}_{\boldsymbol{\phi}}\left(\mathbf{z} \mid \mathbf{x}\right)$:

$$q_{\boldsymbol{\phi}}\left(\mathbf{z} \mid \mathbf{x}\right) \equiv \mathbb{E}_{\boldsymbol{\eta}}\left[\hat{q}_{\boldsymbol{\phi}}\left(\mathbf{z} \mid \mathbf{x}\right)\right], \qquad (22)$$

following Benton et al. [2020], van der Ouderaa and van der Wilk [2022], Immer et al. [2022]. We found that this approach worked well for a single transformation (e.g., rotation) but that it quickly broke down as the space of transformations was expanded (e.g., to all affine transformations; see Figure 13). We hypothesize that the averaging of many latent codes makes it difficult to learn an invariant representation $\mathbf{z}$ without throwing away almost *all* of the information in $\mathbf{x}$. This further motivates our SSL algorithm for learning invariant prototypes. A similar observation was also made by Dubois et al. [2021], who found that an SSL-based objective was superior to an ELBO-based method for learning invariant representations in the context of compression.

## B  Further Practical Considerations

This section elaborates on Section 3.1 and provides additional considerations.

**Suitability of NN architectures.**  The architecture of $f_{\boldsymbol{\omega}}$ must be compatible with learning an equivariant mapping from $\mathbf{x}$ to $\boldsymbol{\eta}$. For example, a standard CNN requires many convolutional filters to represent a function that is (approximately) equivariant to continuous rotations [Maile et al., 2023].

**$\mathcal{X}$-space vs. $\mathcal{H}$-space SSL objective.**  One might notice that it is possible to remove the $\mathcal{T}_{\boldsymbol{\eta}}^{-1}$ operations from both paths of the SSL objective in Figure 4 and still have a valid objective (in $\mathcal{H}$-space rather than $\mathcal{X}$-space). However, the $\mathcal{X}$-space version is preferred since different parameters $\boldsymbol{\eta}_1, \boldsymbol{\eta}_2$ can map to the same transformed element $\mathcal{T}_{\boldsymbol{\eta}_1}(\boldsymbol{x}) = \mathcal{T}_{\boldsymbol{\eta}_2}(\boldsymbol{x})$. E.g., consider rotations transformations applied to various shapes: for a square $\mathcal{T}_{0°} \equiv \mathcal{T}_{90°} \equiv \mathcal{T}_{180°} \equiv \mathcal{T}_{270°}$ all map to the same transformed image, and an $\mathcal{H}$-space objective incorrectly penalizes differences of $\pm n \times 90°$ in $\boldsymbol{\eta}$ values.

We compare rotation inference nets—with hidden layers of dimensions $[2048, 1024, 512, 256, 128]$ trained for 2k steps using the AdamW optimizer with a constant learning rate of $3 \times 10^{-4}$ and a batch size of 256—trained on fully rotated MNIST digits using both $\mathcal{X}$-space and $\mathcal{H}$-space SSL objectives:

| Objective | x-mse | $\eta$-mse |
|---|---|---|
| $\mathcal{X}$-space | **0.2387** | 0.9715 |
| $\mathcal{H}$-space | 0.3567 | 0.4736 |
| average of $\mathcal{X}$-space and $\mathcal{H}$-space | 0.3129 | **0.4619** |

When using the $\mathcal{H}$-space objective, we see the distance in observation ($\mathcal{X}$) space.

**Learning $q_{\boldsymbol{\omega}}(\boldsymbol{\eta}|\mathbf{x})$ instead of $f_{\boldsymbol{\omega}}$.** We found that learning $f_{\boldsymbol{\omega}}$ probabilistically—i.e., allowing for some uncertainty in the transformation during the training process by parameterizing a density over $\mathcal{H}$ with $q_{\boldsymbol{\omega}}(\boldsymbol{\eta}|\mathbf{x})$ and sampling $\boldsymbol{\eta}$—provides small improvements in performance. The distribution $q_{\boldsymbol{\omega}}(\boldsymbol{\eta}|\mathbf{x})$ quickly collapses to a delta. Thus, we hypothesize that the added noise from sampling acts as a regularizer that is helpful at the start of training.

**Inference network blurring schedule.** Occasionally, depending on the dataset, random seed, kind of transformations being applied, and other hyperparameters, training the inference network fails, and the prototype transformations would be 100% lossy—i.e., they would result in completely empty images—regardless of the strength of the invertibility loss. We found that we could prevent this from happening by adding a small amount of Gaussian blur to each example. Furthermore, we found that we only needed to add this blur for a small fraction of the initial training steps to prevent the model from falling into this degenerate local optima.

**Averaging multiple samples for the SSL loss.** Just as we found averaging the MLE loss over multiple samples to improve performance, so too is averaging the SSL loss.

We compare rotation inference nets—with hidden layers of dimensions $[2048, 1024, 512, 256, 128]$ trained for 2k steps using the AdamW optimizer with a cosine decayed with warmup learning rate schedule that starts at $1 \times 10^{-4}$, increases to $3 \times 10^{-4}$ in 500 steps, and then decreases to $1 \times 10^{-7}$, with a batch size of 256—trained on fully rotated MNIST digits using the SSL objective averaged over 1, 3, 5, 10, and 30 samples:

| Samples | x-mse |
|---|---|
| 1 | 0.0981 |
| 3 | 0.0901 |
| 5 | **0.0840** |
| 10 | 0.0853 |
| 30 | 0.0870 |

As the number of samples increases, x-mse decreases until saturating around 5 samples. Note that this relationship is not likely to be *monotonically* decreasing because there is random noise in each training run (i.e., due to random NN initialization, etc.). That said, we expect it will decrease on average as the number of samples increases. We find 5 samples to be a good trade-off between improved performance and increased compute.

**Symmetric SSL loss.** In our SSL loss, based on Figure 4, we are essentially comparing the prototypes given $\boldsymbol{x}$ and $\boldsymbol{x}_{\text{rnd}}$ (a randomly transformed version of $\boldsymbol{x}$). An alternative is to compare the prototypes given $\boldsymbol{x}_{\text{rnd1}}$ and $\boldsymbol{x}_{\text{rnd2}}$, two randomly transformed versions of $\boldsymbol{x}$:

$$\left\| \mathcal{T}^{-1}_{f_{\boldsymbol{\omega}}(\boldsymbol{x}_{\text{rnd1}})}(\boldsymbol{x}_{\text{rnd1}}) - \mathcal{T}^{-1}_{f_{\boldsymbol{\omega}}(\boldsymbol{x}_{\text{rnd2}})}(\boldsymbol{x}_{\text{rnd2}}) \right\|^2_2, \ \boldsymbol{x}_{\text{rnd1}} = \mathcal{T}_{\boldsymbol{\eta}_{\text{rnd1}}}(\boldsymbol{x}), \ \boldsymbol{x}_{\text{rnd2}} = \mathcal{T}_{\boldsymbol{\eta}_{\text{rnd2}}}(\boldsymbol{x}), \ \boldsymbol{\eta}_{\text{rnd1}}, \boldsymbol{\eta}_{\text{rnd2}} \sim p(\boldsymbol{\eta}_{\text{rnd}}).$$
$$(23)$$

As before, we modify this loss to allow us to compose transformations to get

$$\left\| \mathcal{T}_{f_{\boldsymbol{\omega}}(\boldsymbol{x}_{\text{rnd2}})} \circ \mathcal{T}^{-1}_{f_{\boldsymbol{\omega}}(\boldsymbol{x}_{\text{rnd}})}(\boldsymbol{x}_{\text{rnd}}) - \boldsymbol{x}_{\text{rnd2}} \right\|^2_2. \tag{24}$$

The motivation for using this 'symmetric' SSL loss is that it provides the inference network with additional data augmentation—the inference network is now unlikely ever to see the $x$ twice. We find that while this works well for MNIST, it *does not* work well for dSprites. This is because the transformations in dSprites in dSprites are more lossy than those for MNIST. E.g., it is easier to shift a small sprite out of the frame of an image compared to a large digit. Thus, the symmetric loss results in a much higher variance when used with dSprites, which negatively impacts training.

**Composing affine transformations of images.** Care must be taken when composing affine transformations of images when implemented via a coordinate transformation (e.g., `affine_grid` & `affine_sample` in PyTorch, or `scipy.map_coords` in Jax). To compose two affine transformations parameterised by $\eta_1$ and $\eta_2$, the affine matrices $T(\eta_1), T(\eta_2)$ need to be *right*-multiplied with one another; in other words $\mathcal{T}_{\eta_2} \circ \mathcal{T}_{\eta_1} = \mathcal{T}'_{T(\eta_1)T(\eta_2)}$. This is because, in these implementations of affine transformation of images, the affine transformation is applied to the pixel grid (i.e., the reference frame), rather than to the image itself. In effect, the resulting transformation as applied to the objects in the image is the opposite; if the reference frame moves to the right, the objects in the image move to the left, etc. More concretely, when the reference frame is affine-transformed by $\mathcal{T}$, the image itself is affine-transformed by $\mathcal{T}^{-1}$.

**Overfitting of the generative network.** While we did not observe any overfitting of the inference network (likely due to the built-in 'data augmentation' of our SSL loss, and the general difficulty of learning a function with equivariance to arbitrary transformations), we did find that the generative network was prone to overfitting. We addressed this by using a validation set to optimize several relevant hyper-parameters (e.g., dropout rates, number of flow layers, number of training epochs, etc.); see Appendix C.

**Learning $p_\psi(\eta \,|\, \hat{x})$ with imperfect inference, continued.** To encourage $p_\psi(\eta \,|\, \hat{x})$ produce the same distribution for the inconsistent prototypes produced by $q_\omega(\eta \,|\, x)$, we add a *consistency* loss to line 8 of Algorithm 1 the MLE objective:

$$L_{\text{consistency}}(\psi) = \frac{1}{N^2} \sum_{i=1}^{N} \sum_{j=1}^{N} |\log p_i - \log p_j|, \tag{25}$$

where $p_i = p_\psi(\eta_x \,|\, \hat{x}'_i)$ and $\hat{x}'_i$ is due to the $i^{\text{th}}$ $\eta_{\text{rnd}}$ sample.

## C Experimental Setup

We use `jax` with `flax` for NNs, `distrax` for probability distributions, and `optax` for optimizers. We use `ciclo` with `clu` to manage our training loops, `ml_collections` to specify our configurations, and `wandb` to track our experiments. The code is available at `https://github.com/cambridge-mlg/sgm`.

Unless otherwise specified, we use the following NN architectures and other hyperparameters for all of our experiments. We use the AdamW optimizer with weight decay of $1 \times 10^{-4}$, global norm gradient clipping, and a linear warm-up followed by a cosine decay as a learning rate schedule. The exact learning rates and schedules for each model are discussed below. We use a batch size of 512.

All of our MLPs use `gelu` activations and `LayerNorm`. In some cases, we use `Dropout`. The structure of each layer is `Dense → gelu → LayerNorm → Dropout`. Whenever we learn or predict a scale parameter $\sigma$, it is constrained to be positive using a `softplus` operation.

**Inference network.** We use a MLP with hidden layers of dimension $[2048, 1024, 512, 256]$. The network outputs a mean $\eta$ prediction for each example and the uncertainty—as mentioned in Appendix B—is implemented as a homoscedastic scale parameter. We train for 60k steps. For each example, we average the loss over 5 random augmentations. In some settings—also mentioned in Appendix B—we add a small amount of blur to the images with a Gaussian filter of size 5 for the first 1% of training steps. The $\sigma$ value for the filter was linearly decayed from their maximum to 0. The initial maximum value is specified below.

**Generative network.** Our generative model is a Neural Spline Flow [Durkan et al., 2019] with 6 bins in the range $[-3, 3]$. We use an MLP with hidden layers of dimension $[1024, 512, 512]$ as a shared feature extractor. The base normal distribution's mean and scale parameters are predicted by another MLP, with hidden layers of dimension $[256, 256]$, whose input is the shared feature representation. The parameters of the spline at each layer of the flow are predicted by MLPs with a single hidden layer of dimension 256, with a dropout rate of 0.1, whose input is a concatenation of the shared feature representation, and the (masked) outputs of the previous layer. For each example, we average the loss over 5 random augmentations.

## C.1 MNIST under affine transformations

We make use of the MNIST dataset [LeCun et al., 2010], which is available under the MIT license.

We split the MNIST training set by removing the last 10k examples and using them exclusively for validation and hyperparameter sweeps.

When randomly augmenting the inputs for our SSL (see Section 2.1 and Figure 4) and MLE (see Section 3.1) losses, we sample transformation parameters from $\mathcal{U}(-\boldsymbol{\eta}_{\max}, \boldsymbol{\eta}_{\max})$, where $\boldsymbol{\eta}_{\max} = (0.25, 0.25, \pi, 0.25, 0.25)$ is the maximum ($x$-shift, $y$-shift, rotation, $x$-scale, $y$-scale) applied to the images. All affine transformations are applied with bi-cubic interpolation.

**Inference network.** The invertibility loss $\mathcal{L}_{\text{invertibility}}$ (7) is multiplied by a factor of 0.1. For the VAE data-efficiency results in Figure 11, we performed the following hyperparameter grid search for each random seed and amount of training data:

- blur $\sigma_{\text{init}} \in [0, 3]$,
- gradient clipping norm $\in [3, 10]$,
- learning rate $\in [1{\times}10^{-3}, 3{\times}10^{-4}, 1{\times}10^{-4}]$,
- initial learning rate multiplier $\in [3{\times}10^{-2}, 1{\times}10^{-2}]$,
- final learning rate multiplier $\in [1{\times}10^{-3}, 3{\times}10^{-4},]$, and
- warm-up steps % $\in [0.05, 0.1, 0.2]$.

All of the other MNIST affine transformation results use a blur $\sigma_{\text{init}}$ of 0, a gradient clipping norm of 10, a learning rate of $3{\times}10^{-4}$, an initial learning rate multiplier of $1{\times}10^{-2}$, a final learning rate multiplier of $1{\times}10^{-3}$, and a warm-up steps % of 0.2, which are the best hyperparameters for 50k training examples with an arbitrarily chosen random seed. We use the 'symmetric' SLL loss discussed in Appendix B.

**Generative network.** We use an initial learning rate multiplier of 0.1, a gradient clipping norm of 2, and a warm-up steps % of 0.2. For the VAE data-efficiency results in Figure 11, we performed the following hyperparameter grid search for each random seed and amount of training data:

- learning rate $\in [3{\times}10^{-3}, 3{\times}10^{-4}]$,
- final learning rate multiplier $\in [0.3, 0.03]$,
- number of training steps $\in [7.5\text{k}, 15\text{k}, 30\text{k}, 60\text{k}]$,
- number of flow layers $\in [4, 5, 6]$,
- shared feature extractor dropout rate $\in [0.05, 0.1, 0.2]$, and
- consistency loss multiplier $\in [0, 1]$ (whether or not to use (25)).

Note that we use the log-likelihood of the validation data under the generative model to select the best hyper-parameters. I.e., we do not use the total loss, which may or may not include the consistency term, since these losses are not directly comparable. We require a trained inference network when sweeping over the generative network hyperparameters. We use the inference network hyperparameters for the same (random seed, number of training examples) pair. All of the other MNIST affine transformation results use a learning rate of $3{\times}10^{-3}$, a final learning rate multiplier of 0.03, 60k training steps, 6 flow layers, a dropout rate of 0.2 in the shared feature extractor, and a consistency loss multiplier of 1, which are the best hyperparameters for 50k training examples.

## C.2 MNIST under color transformations

We follow the same setup as above for color transformation on the MNIST dataset, with the following exceptions. We do not use an invertibility loss when training the inference network. Instead, for both the inference and generative networks, we constrain the outputs to be in $[-\boldsymbol{\eta}_{\max}, \boldsymbol{\eta}_{\max}] + (0.5, 0., 0.)$, where $\boldsymbol{\eta}_{\max} = (0.5, 2.301, 0.51)$ using with `tanh` and `scale` bijectors. We randomly augment the inputs by sampling transformation parameters from $\mathcal{U}(-\boldsymbol{\eta}_{\max} + (0.5, 0., 0.), \boldsymbol{\eta}_{\max} + (0.5, 0., 0.))$.

**Inference network.** We use a blur $\sigma_{\text{init}}$ of 3, a gradient clipping norm of 2, a learning rate of $3\times10^{-4}$, an initial learning rate multiplier of $1\times10^{-2}$, a final learning rate multiplier of $1\times10^{-4}$, and a warm-up steps % of 0.1, which were chosen using the same grid sweep as MNIST with affine transformations.

**Generative network.** We use a learning rate of $3\times10^{-3}$, with an initial learning rate multiplier of $1\times10^{-1}$, a final learning rate multiplier of $3\times10^{-2}$, 15k training steps, 6 flow layers, and a dropout rate of 0.2 in the shared feature extractor.

## C.3 dSprites under affine transformations

We make use of the dSprites dataset [Matthey et al., 2017], which is available under the Apache 2.0 license.

For our dSprites experiments, we follow the same setup as for MNIST under affine transformations above, with the following exceptions. We do not use an invertibility loss when training the inference network. Instead, for both the inference and generative networks, we constrain their outputs to be in $[-\boldsymbol{\eta}_{\max}, \boldsymbol{\eta}_{\max}]$, where $\boldsymbol{\eta}_{\max} = (0.75, 0.75, \pi, 0.75, 0.75)$ using with `tanh` and `scale` bijectors. We *do not* use the 'symmetric' SSL loss discussed in Appendix B.

**Inference network.** We randomly augment the inputs by sampling transformation parameters from $\mathcal{U}(-\boldsymbol{\eta}_{\max}, \boldsymbol{\eta}_{\max})$, where $\boldsymbol{\eta}_{\max}$ matches the constraints above. We use a blur $\sigma_{\text{init}}$ of 3, a gradient clipping norm of 3, a learning rate of $1\times10^{-3}$, an initial learning rate multiplier of $3\times10^{-2}$, a final learning rate multiplier of $1\times10^{-3}$, and a warm-up steps % of 0.05, which were chosen using the same grid sweep as MNIST with affine transformations.

**Generative network.** We randomly augment the inputs by sampling transformation parameters from $\mathcal{U}(-\boldsymbol{\eta}_{\max} \times 0.75, \boldsymbol{\eta}_{\max} \times 0.75)$, where $\boldsymbol{\eta}_{\max}$ matches the constraints above. We use a learning rate of $3\times10^{-4}$, a final learning rate multiplier of 0.3, 60k training steps, 6 flow layers, and a dropout rate of 0.05 in the shared feature extractor, which were chosen using the same grid sweep as MNIST with affine transformations.

Although we swept over the consistency loss multiplier, we accidentally always used a consistency loss multiplier of 1 in our experiments. This means that for some (random seed, amount of training data) pairs the performance of our generative network is slightly lower than it should be since the chosen hyperparameters may correspond to a consistency loss multiplier of 0. We include this detail for reproducibility but note that it does not change our findings in any material way.

### C.3.1 dSprites Setup

The original dSprites dataset contains sprites with the following factors of variation [Matthey et al., 2017].

- Color: white
- Shape: square, ellipse, heart
- Scale: 6 values linearly spaced in $[0.5, 1]$
- Orientation: 40 values linearly spaced in $[0, 2\pi]$
- X position: 32 values linearly spaced in $[0, 1]$
- Y position: 32 values linearly spaced in $[0, 1]$

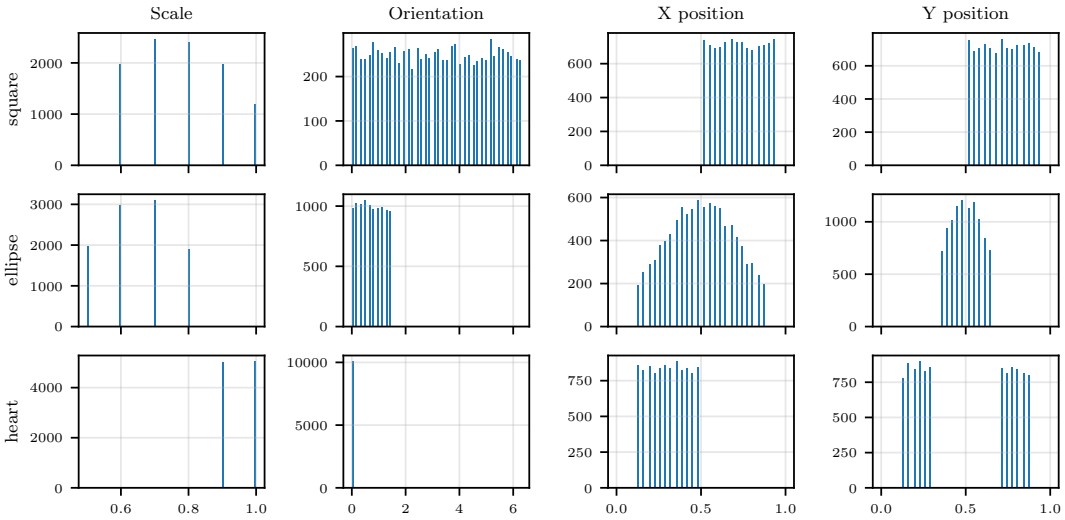

Figure 14: Latent factor distributions for our modified dSprites data loader.

The dataset consists of sprites with the outer product of these factors, for a total of 737280 examples. We modified our data loader to resample the sprites proportional to the following distributions on the latent factors conditioned on the shape.

- **square**
    - Scale: TruncNorm $\left( \mu = 0.75,\ \sigma^2 = 0.2,\ \min = 0.55,\ \max = 1.0 \right)$
    - Orientation: $\mathcal{U}(0.0,\ 2\pi)$
    - X position: $\mathcal{U}(0.5,\ 0.95)$
    - Y position: $\mathcal{U}(0.5,\ 0.95)$

- **ellipse**
    - Scale: TruncNorm $(0.65,\ 0.15,\ 0.5,\ 0.85)$
    - Orientation: $\mathcal{U}(0.0,\ \pi/2)$
    - X position: TruncNorm $(0.5,\ 0.25,\ 0.1,\ 0.9)$
    - Y position: TruncNorm $(0.5,\ 0.15,\ 0.35,\ 0.65)$

- **heart**
    - Scale: $\mathcal{U}(0.9,\ 1.0)$
    - Orientation: $\delta(0.0)$
    - X position: $\mathcal{U}(0.1,\ 0.5)$
    - Y position: $0.5 \cdot \mathcal{U}(0.1,\ 0.3) + 0.5 \cdot \mathcal{U}(0.7,\ 0.9)$

An example of the resulting empirical distributions over the latent factors is shown in Figure 14. The three shapes are sampled with equal proportions.

### C.4 GalaxyMNIST under affine and color transformations

We make use of the GalaxyMNIST dataset [Walmsley et al., 2022], which is available under the GPL-3.0 licence.

For our GalaxyMNIST experiments, we follow the same setup as for MNIST under affine transformations above, with the following exceptions. We do not use an invertibility loss when training the inference network. Instead, for both the inference and generative networks, we constrain their outputs to be in $[-\boldsymbol{\eta}_{\max},\ \boldsymbol{\eta}_{\max}] + (0., 0., 0., 0., 0., 0.5, 0., 0.)$, where $\boldsymbol{\eta}_{\max} = (0.75, 0.75, \pi, 0.75, 0.75, 0.5, 2.31, 0.51)$ using with `tanh` and `scale` bijectors. This dataset contains 10k examples. We use the last 2k as our test set, and the previous 1k as a validation set.

**Inference network.** We use a MLP with hidden layers of dimension $[1024, 1024, 512, 256]$. We train for 10k steps. We randomly augment the inputs by sampling transformation parameters from $\mathcal{U}(-\boldsymbol{\eta}_{\max} + (0., 0., 0., 0., 0., 0.5, 0., 0.), \boldsymbol{\eta}_{\max} + (0., 0., 0., 0., 0., 0.5, 0., 0.))$, where $\boldsymbol{\eta}_{\max}$ matches the constraints above. For the VAE data-efficiency results in Figure 12, we performed the same hyperparameter grid search as above for each random seed and amount of training data. All of the other GalaxyMNIST results use a blur $\sigma_{\text{init}}$ of 0, a gradient clipping norm of 10, a learning rate of $3 \times 10^{-4}$, an initial learning rate multiplier of $1 \times 10^{-2}$, a final learning rate multiplier of $3 \times 10^{-4}$, and a warm-up steps % of 0.2, which are the best hyperparameters for 7k training examples with an arbitrarily chosen random seed. We use the 'symmetric' SLL loss discussed in Appendix B.

**Generative network.** We randomly augment the inputs by sampling transformation parameters from $\mathcal{U}(-\boldsymbol{\eta}_{\max} \times 0.75 + (0., 0., 0., 0., 0., 0.5, 0., 0.), \boldsymbol{\eta}_{\max} \times 0.75 + (0., 0., 0., 0., 0., 0.5, 0., 0.))$, where $\boldsymbol{\eta}_{\max}$ matches the constraints above. For the VAE data-efficiency results in Figure 12, we perform the same hyperparameter grid search as above for each random seed and amount of training data, with the following changes.[7] The sweep for number of training steps is $[3.75k, 7.5k, 15k]$. All of the other GalaxyMNIST results use a learning rate of $3 \times 10^{-4}$, a final learning rate multiplier of 0.03, 15k training steps, 4 flow layers, a dropout rate of 0.05 in the shared feature extractor, and a consistency loss multiplier of 1, which were chosen using the same grid sweep for an arbitrary random seed and 7k training examples.

## C.5 PatchCamelyon under affine and color transformations

We make use of the PatchCamelyon dataset [Veeling et al., 2018], which is available under the Creative Commons Zero v1.0 Universal license.

We resized the images from $96 \times 96$ pixels to $64 \times 64$ using bilinear interpolation. The dataset has dedicated train, test, and validation splits which we use without any modifications.

We follow the same setup as for GalaxyMNIST under affine and color transformations above, with the exceptions listed below. We only used a single random seed.

**Inference network.** We train for 20k steps.

**Generative network.** The sweep for number of training steps is $[15k, 30k, 60k]$.[8]

## C.6 VAE, AugVAE, and InvVAE

Our VAEs use a latent code size of 20. The prior is a normal distribution with learnable mean and scale, initialized to 0s and 1s, respectively.

Our VAE encoders are LeNet-style CNNs with convolutional feature extractors followed by an MLP with a single hidden layer of size 256. The convolutional feature extractors use `gelu` activations and `LayerNorm`. The structure is `Conv` $\rightarrow$ `gelu` $\rightarrow$ `LayerNorm`. All `Conv` layers use $3 \times 3$ filters. The first two `Conv` have a stride of 2, while all others have a stride of 1. In between the convolutional layers and the MLP, there is a special dimensionality reduction `Conv` with only 3 filters followed by a `flatten`. For each dimension of the latent code, the encoder predicts a mean $\mu$ and a scale $\sigma$. The means and scales are initialized to 0s and 1s, respectively.

Our VAE decoders are inverted versions of our encoders. That is, we reverse the order of all of the `Dense` and `Conv` layers. The dimensionality reduction `Conv` layer and the `flatten` operation are replaced with the appropriate `Dense` layer and `reshape` operation. We replace all other `Conv` layers with `ConvTransposed` layers For each pixel of an image, the decoder predicts a mean $\mu$. We learn a homoscedastic per-pixel scale $\sigma$. The scales are initialized to 1.

---

[7] Our GalaxyMNIST results have the same issue as our dSprites results—the sweep included a consistency loss multiplier which was always set to a value of 1 in our experiments. This results in some small performance degradations.

[8] Our PatchCamelyon results have the same consistency multiplier issue as our dSprites and GalaxyMNIST results.

We use an initial learning rate multiplier of $3\times10^{-2}$, and a final learning rate multiplier of $1\times10^{-4}$. We run the following grid sweep for each (random seed, number of training examples, maximum added rotation angle) triplet:

- learning rate $\in [3\times10^{-3}, 6\times10^{-3}, 9\times10^{-3}]$,
- convolutional filters $\in [(64, 128), (64, 128, 256)]$,
- number of training steps $\in [5k, 10k, 20k]$, and
- warm-up steps $\% \in [0.15, 0.2]$.

When running the sweep for AugVAE and InvVAE, we use the inference and generative network hyperparameters for the same (random seed, number of training examples) pair.

### C.6.1   PatchCamelyon

For our PatchCamelyon experiments, we use only a single random seed and a slightly modified hyperparameter sweep:

- learning rate $\in [3\times10^{-3}, 6\times10^{-3}$,
- convolutional filters $\in [(64, 128), (64, 128, 256), (128, 256, 512)]$,
- number of dense hidden layers $\in [1, 2]$,
- number of training steps $\in [20k, 30k, 40k]$, and
- warm-up steps $\% \in [0.15]$.

### C.7   Parametrisations of Symmetry transformations

We consider five affine transformations: shift in $x$, shift in $y$, rotation, scaling in $x$, and scaling in $y$. We represent these transformations using affine transformation matrices $\boldsymbol{A} = \exp\left(\sum_i \eta_i \boldsymbol{G}_i\right)$, where $\boldsymbol{G}_i$ are generator matrices for rotation, translation, and scaling; see Benton et al. [2020]. The transformations are applied to an image by transforming the coordinates $(x, y)$ of each pixel, as in Jaderberg et al. [2015]: $\begin{bmatrix} x' & y' & 1 \end{bmatrix}^\mathsf{T} = \boldsymbol{A} \cdot \begin{bmatrix} x & y & 1 \end{bmatrix}^\mathsf{T}$.

To parameterize color transformations, we use an equivalent representation of color images in Hue-Saturation-Value (HSV) space, where each pixel is represented as a tuple $(h, s, v) \in \{[-\pi, \pi] \times [0, 1] \times [0, 1]\}$. Intuitively, HSV space represents the color of each pixel in a conical space where the hue corresponds to the rotation angle around the cone's vertical axis, the saturation corresponds to the radial distance from the cone's center, and the value corresponds to the distance along the cone's vertical axis, with a value of 0 corresponding to the tip of the cone, and a value of 1 corresponding to the base of the cone. We color-transform an image by transforming each pixel as

$$\begin{bmatrix} h' \\ s' \\ v' \end{bmatrix} = \begin{bmatrix} (h + 2\pi\eta_h) \mod 2\pi \\ \max(0, \min(s \exp(\eta_s), 1)) \\ \max(0, \min(v \exp(\eta_v), 1)) \end{bmatrix}. \tag{26}$$

We therefore obtain $\boldsymbol{\eta} = (\eta_h, \eta_s, \eta_v) \in \{[0, 1] \times \mathbb{R} \times \mathbb{R}\}$. We choose this specific form of parametrizing the $\boldsymbol{\eta}$ parameters in order to gain the convenience of simply adding and subtracting in $\boldsymbol{\eta}$ space when carrying out color transform compositions and inverses. More concretely, with our chosen parametrization, we obtain the property that $\mathcal{T}_{\boldsymbol{\eta}_1} \circ \mathcal{T}_{\boldsymbol{\eta}_2} = \mathcal{T}_{\boldsymbol{\eta}_1 + \boldsymbol{\eta}_2}$. Therefore, we can easily perform compositions and inversions in $\boldsymbol{\eta}$ space for color transformations without resorting to matrix multiplications. In order to achieve this, we first consider hue, which is easy to parametrize in an additive fashion using a modulo operation due to the fact that hue is represented as a rotation angle in HSV space. On the other hand, saturation and value are discontinuous parameters that vary between 0 and 1, and cannot be directly modeled in an additive fashion, as they can't take values outside their range. Instead, we model them as multiplicative factors in $\mathbb{R}^+$, where we first exponentiate $\eta_s$ and $\eta_v$ to ensure the multiplicative factors are positive. We further clip the obtained values to ensure they are in the range $[0, 1]$. This parametrization allows us to effectively add parameters to compose them, as the multiplicative factors compose in exponent space.

In order to ensure that we can easily backpropagate through the clipping operation, we define a `passthrough_clip` function in Jax, where we define a custom gradient that doesn't zero out gradients even if the inputs to the function are out of bounds. We find that using the `passthrough_clip` operation is essential to training the model.

# D  Compute Requirements

The experiments for this paper were performed on a cluster equipped with NVIDIA A100 GPUs. All model training requires only a single such GPU. However, we used up to 64 GPUs at a time to run our hyper-parameter searches in parallel. Including exploratory experiments, all hyperparameter sweeps, discarded runs, etc., *the total compute used for this paper is approximately 250 A100 GPU days.* **The total cost to reproduce the experiments in the paper is approximately 135 A100 GPU days.** We break this cost down as follows. Note that the cost for different figures do not naively sum as hyper-parameter sweeps for some figures are reused for others, as discussed in Appendix C.

**Figure 8a:** 6 days
> **Inference net sweeps:** 4 days
> **Generative net sweeps:** 2 days

**Figure 8b:** 3 days
> **Inference net sweeps:** 2 days
> **Generative net sweeps:** 1 day

**Figure 8c:** 3 days
> **Inference net sweeps:** 2 days
> **Generative net sweeps:** 1 day

**Figure 8d:** 7 days
> **Inference net sweeps:** 6 days
> **Generative net sweeps:** 1 day

**Figure 9:** 3 days
> **Inference net sweeps:** 2 days
> **Generative net sweeps:** 1 day

**Figure 10:** 2 days
> **Inference net sweeps:** 2 days

**Figure 11:** 69 days
> **Inference net sweeps:** 30 days
> **Generative net sweeps:** 12 days
> **VAE sweeps:** 27 days

**Figure 12:** 53 days
> **Inference net sweeps:** 36 days
> **Generative net sweeps:** 8 days
> **VAE sweeps:** 9 days

# E  Additional Results

## E.1  Comparisons to LieGAN

In this section, we compare the ability of our method to learn symmetries to LieGAN [Yang et al., 2023], which uses a generator-discriminator framework to automatically discover equivariances from a dataset using generative adversarial training. Similar to [Yang et al., 2023], we transform the MNIST dataset to have rotations in the range $[-45°, 45°]$, which ensures the dataset contains SE(2) symmetry (rotations and translations). The dataset is processed and our method is trained as described in Section 4.1. For LieGAN, following the experimental design of [Yang et al., 2023], we set the number of generator channels to $c = 1$, and consider learnable 6-dimensional Lie matrices in the generator model. The discriminator model consists of a pre-trained LeNet5 feature extractor as the backbone, and the validator is a 3-layer MLP with 512 hidden units and ReLU activations. We train the GAN for 100 epochs with a batch size of 64, and obtain the Lie matrix below

$$L = \begin{bmatrix} 0.02 & -0.34 & 0.28 \\ 0.33 & 0.08 & -0.05 \\ 0 & 0 & 0 \end{bmatrix}.$$

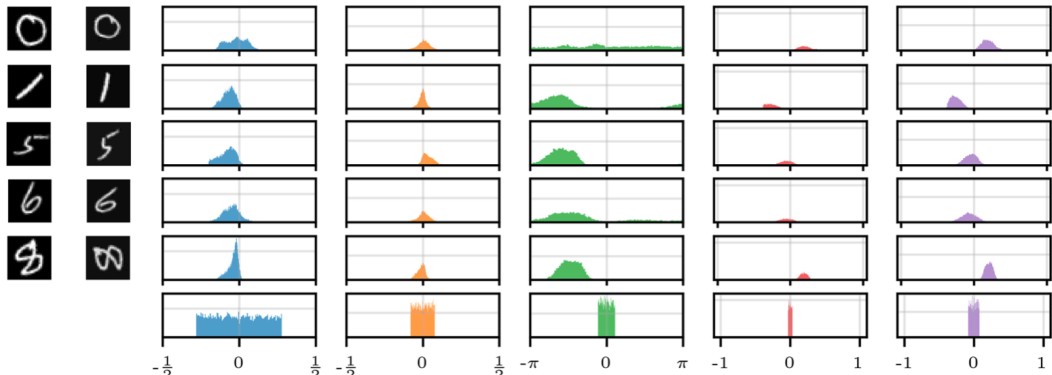

Figure 15: **Learnt augmentation distribution for the MNIST dataset rotated in the range** $[-45°, 45°]$ **for our SGM model, and the LieGAN method. The columns correspond to distributions for translation in** $x$**, translation in** $y$**, rotation, scaling in** $x$**, and scaling in** $y$**. (Row 1-5)** Our SGM learns accurate ranges of rotational invariance present in the training dataset of a width of $\pi/2$ for most training examples, along with learning the natural invariances present in the training data for translations and scaling. Furthermore, for certain digits (i.e. 0), the SGM model accurately predicts a uniform distribution from $[-\pi, \pi]$, signifying that rotationally invariant digits such as a 0 would not display a more narrow rotational invariance. **(Row 6)** On the other hand, the LieGAN model learns a single Lie matrix across the entire training dataset that encodes the maximum possible range of transformations, and predicts a uniform distribution between those ranges. It can be seen that LieGAN inaccurately predicts a large range for translations in $x$, and does not recover the correct range of rotational invariances present in the training dataset.

In Figure 15, we can see that LieGAN struggles to correctly recover the range of invariances present in the training dataset, especially for translations in $x$. It is also unable to provide a fine-grained representation of invariances depending on specific examples or type of digits. We note that we re-implemented the rotated MNIST experiment from Yang et al. [2023], as the code for the image domain experiments was not open-source. Hence, the choice of using a pre-trained LeNet5 model for the discriminator, and the specific hyperparameter configurations, were informed decisions made by us based on ablations. However, our results appear to be inline with those presented by Yang et al. [2023]; concretely, we note that the results presented in their paper also display a mismatch between the invariances present in the dataset and those learned by LieGAN. For example, in their Figure 11, we see that the sampled digits are often rotated by significantly more than 45°. Furthermore, we see evidence of typical GAN mode collapse, with many very similar rotations for each digit.

### E.2  PatchCamelyon — Boundary Effects

In this section, we provide a "negative" result for our SGM when applied to the PatchCamelyon dataset [Veeling et al., 2018]. The examples in this dataset, unlike those used in Section 4, contain "content" up to the boundaries of the images.

Figure 16 shows examples of the prototypes and learned distributions for this dataset, with affine and color transformations. In particular, the allowed rotation was between $\pm 180°$, while the actual dataset has only a rotational invariance of $\pm n \times 90°$. We see that in some cases the prototypes are rotated by close to $\pm n \times 45°$ relative to the original images. In other cases, the rotation of the prototypes relative to the original images is closer to $\pm n \times 90°$. In the latter case, the learned distribution over rotation is close to the true distribution, but in the former case, the model learns a distribution that is closer to uniform. As a result, the resampled digits often display boundary effects that are not present in the original dataset. Otherwise, our SGM has learned reasonable distributions for translation, scaling, and HSV transformations.

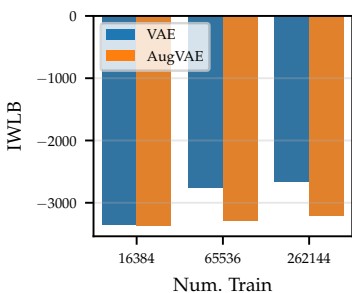

Figure 17: VAE data-efficiency for PatchCamelyon.

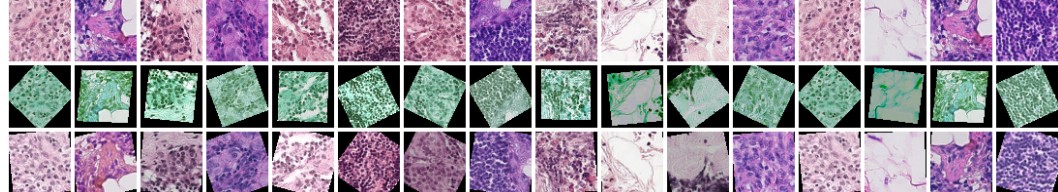

(a) **Top:** samples from the test set. **Mid:** prototypes for each test example. **Bot:** resampled versions of each test example given the prototype.

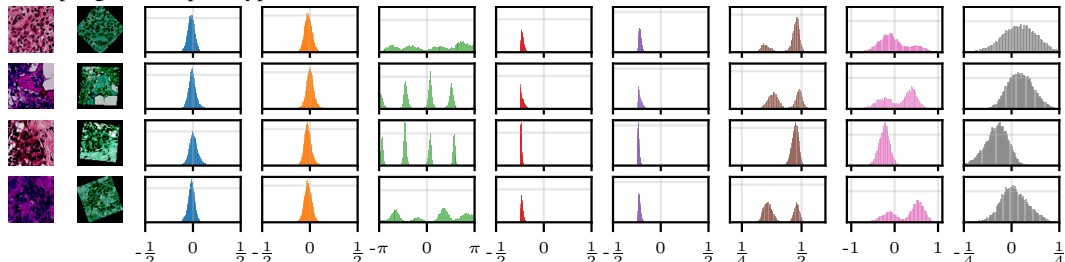

(b) From left to right, test examples, their prototypes, and the corresponding marginal distributions over translation in $x$, translation in $y$, rotation, scaling in $x$, scaling in $y$, hue, saturation, and value.

Figure 16: Prototypes and learned distributions for PatchCamelyon.

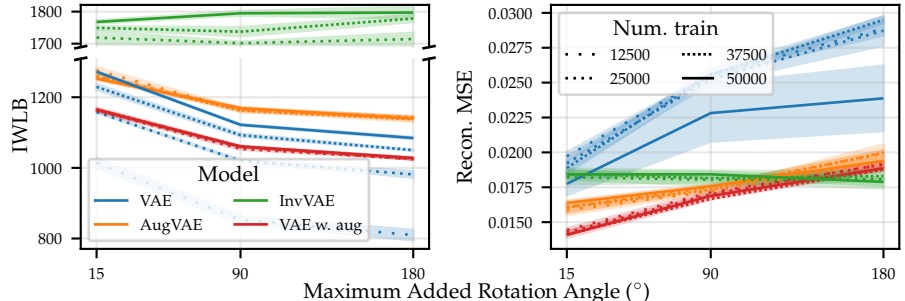

Figure 18: Incorporating symmetries improves data efficiency. Importance-weighted lower bound (IWLB) and reconstruction MSE (mean and std. err. over 3 random seeds) for rotated MNIST with a standard VAE (with and without standard data augmentations) and two VAE variants that incorporate symmetries via our SGM. Improved data efficiency is demonstrated by better performance with less training data, and reduced sensitivity to added rotations.

Figure 17 compares a standard VAE with AugVAE, an SGM-VAE hybrid model. We see that for small amounts of data, the VAE and AugVAE perform similarly. However, as the amount of training data increases, the VAE performs better. This is likely because the SGM has not learned the true distribution over rotations.

This "negative" result highlights the importance of correctly choosing the prior transformation distributions in some settings. In this case, the performance of the SGM would have been improved by choosing a categorical distribution over rotations.

### E.3    Additional Experiments

In this section, we provide additional plots to supplement those in Section 4.

Figure 18 extends the results in Figure 11 to by including an additional metric: reconstruction MSE. Our findings with IWLB are consistent for this metric.

Figure 19 expands on Figure 8b in two ways. Firstly, it makes it clear that our inference network is able to provide the same or very similar prototype for observations in the same orbit. Secondly, it provides many more resampled examples of each digit, further demonstrating that our SGM has correctly captured the symmetries present in the dataset. Figure 20 expands on Figure 8c in the same way.

Figure 21 extends Figure 9 by including all of the digits shown in Figure 19. The conclusions are much the same as before. We see that the learned distributions all make sense, especially for the most easily interpretable transformation parameter, rotations. Again, we note that smaller and bigger prototypes have appropriately different scaling distributions. Figure 22 provides the learnt marginal distributions for the digits in Figure 20. Here, we manually controlled the distributions over hue and saturation when loading the dataset, so we know that the range of the hue distribution should be approximately $\pi$, while the range of the saturation distribution should be around $0.3$. We see that this is indeed the case. We did not control the value of the images, so it is more difficult to interpret those. However, given that most (non-black) pixels are bright (i.e., close to 1) it makes sense that our SGM learns multiplicative values closer to 1.

Finally, Figure 23 extends our dSprites results in two ways. Firstly, it provides many more resampled sprites, which also serves to demonstrate further that our SGM has captured the symmetries correctly. Secondly, the figure includes empirical distributions of positions of each of the classes of digits, which we have carefully controlled as described in Appendix C.3.1. These empirical distributions for the dataset are compared with empirical distributions for our resampled sprites. We see that although the resampled densities don't match the original densities perfectly, their general shapes and ranges are correct.

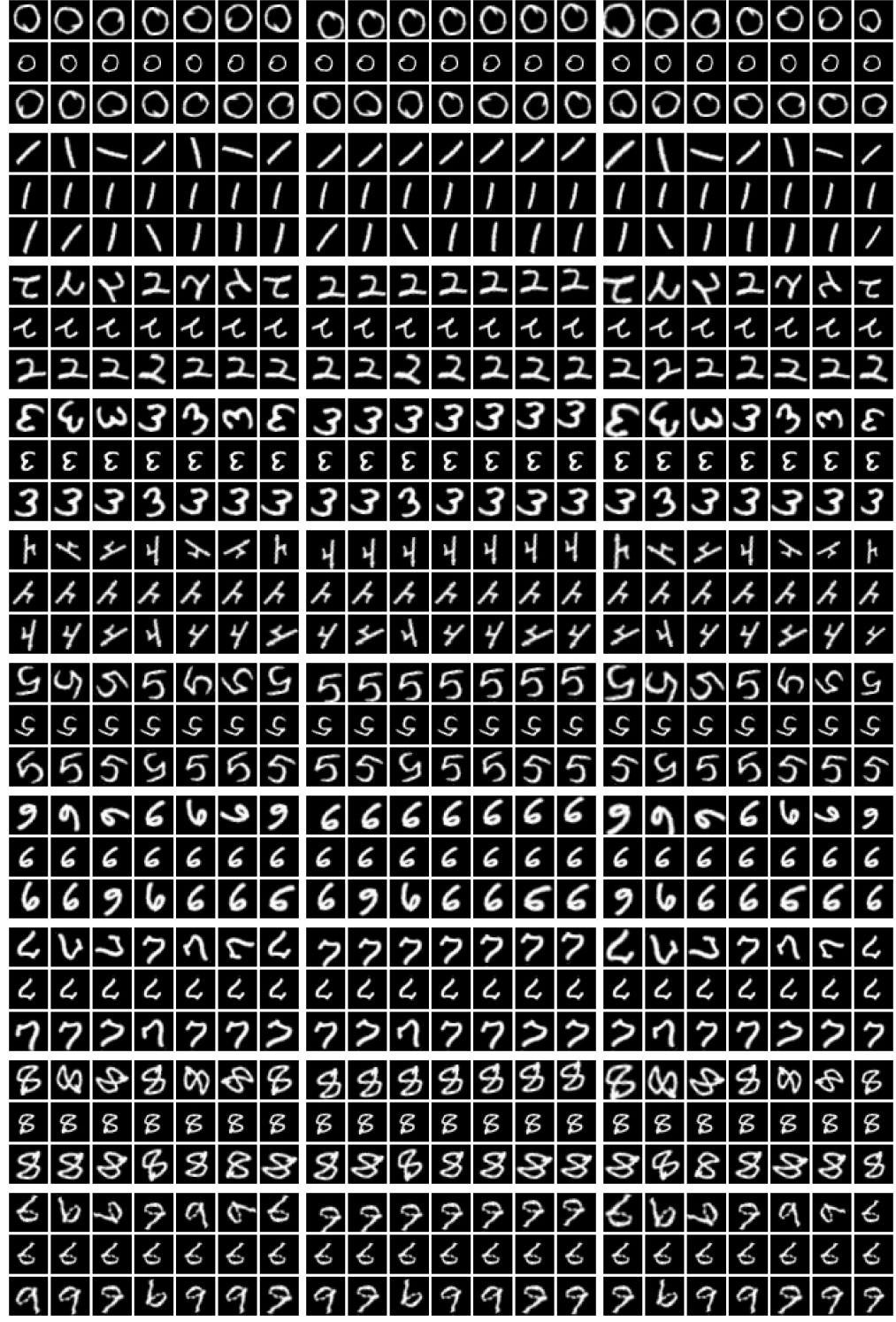

Figure 19: Columns from left to right: only rotation, only translations, translation + rotation + scaling. Each of the blocks in this figure follows the same format. **Top:** 7 examples from the same orbit. **Mid:** The corresponding prototypes. **Bot:** Resampled versions of the digits, given the prototypes.

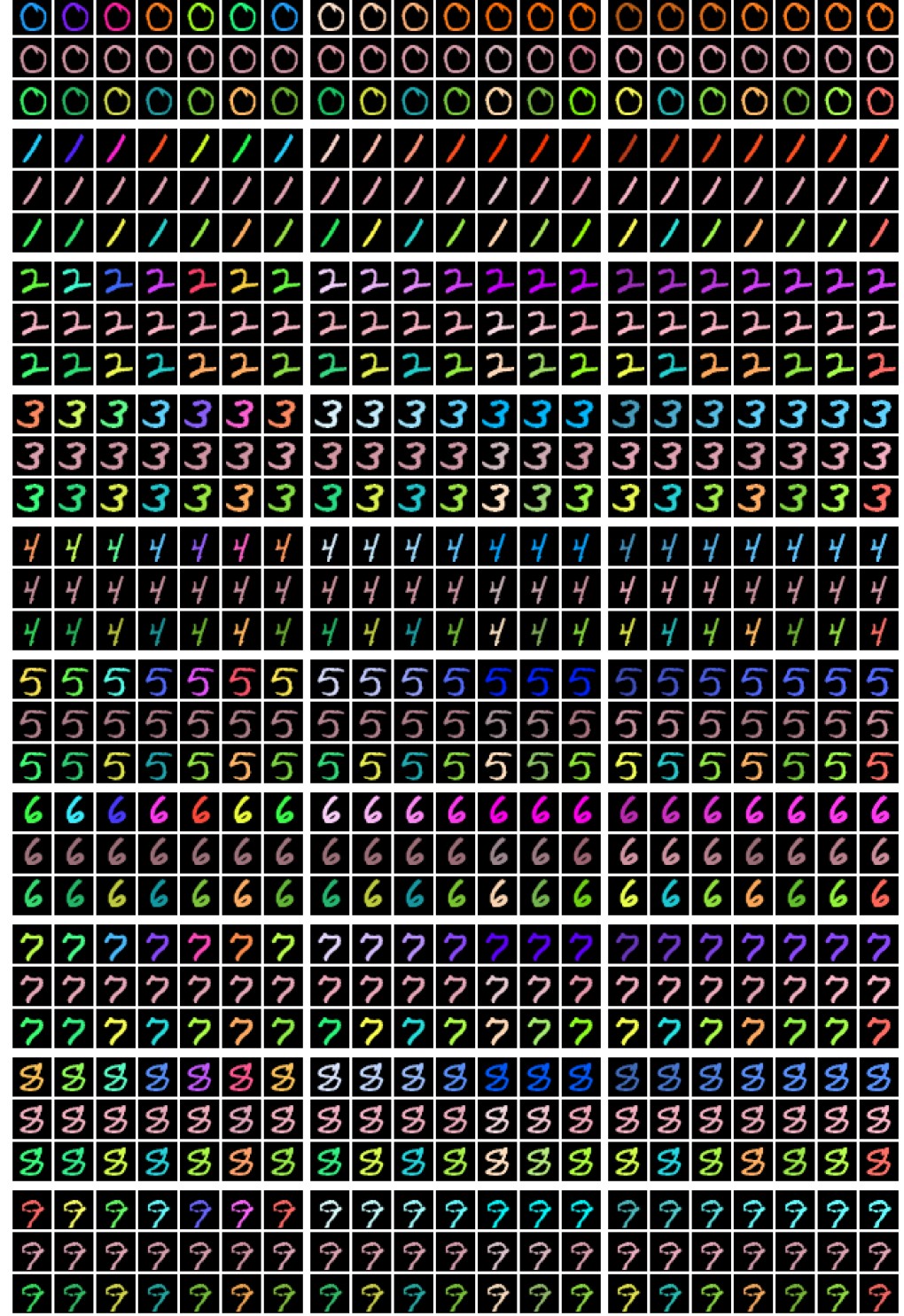

Figure 20: Columns from left to right: only hue, only saturation, only value. Each of the blocks in this figure follows the same format. **Top:** 7 examples from the same orbit. **Mid:** The corresponding prototypes. **Bot:** Resampled versions of the digits, given the prototypes.

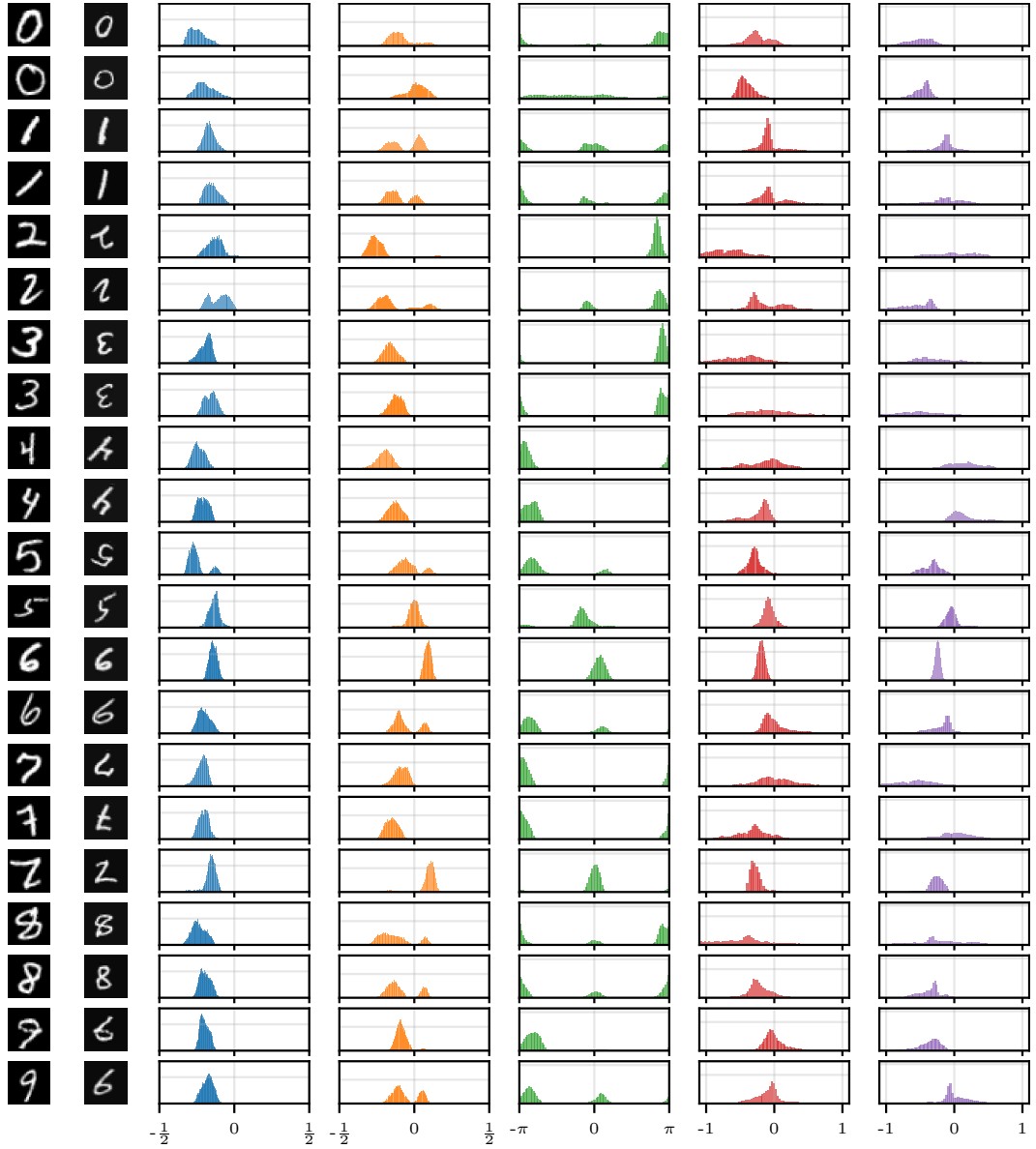

Figure 21: From left to right, test examples from MNIST, their prototypes, and the corresponding marginal distributions over translation in $x$, translation in $y$, rotation, scaling in $x$, and scaling in $y$.

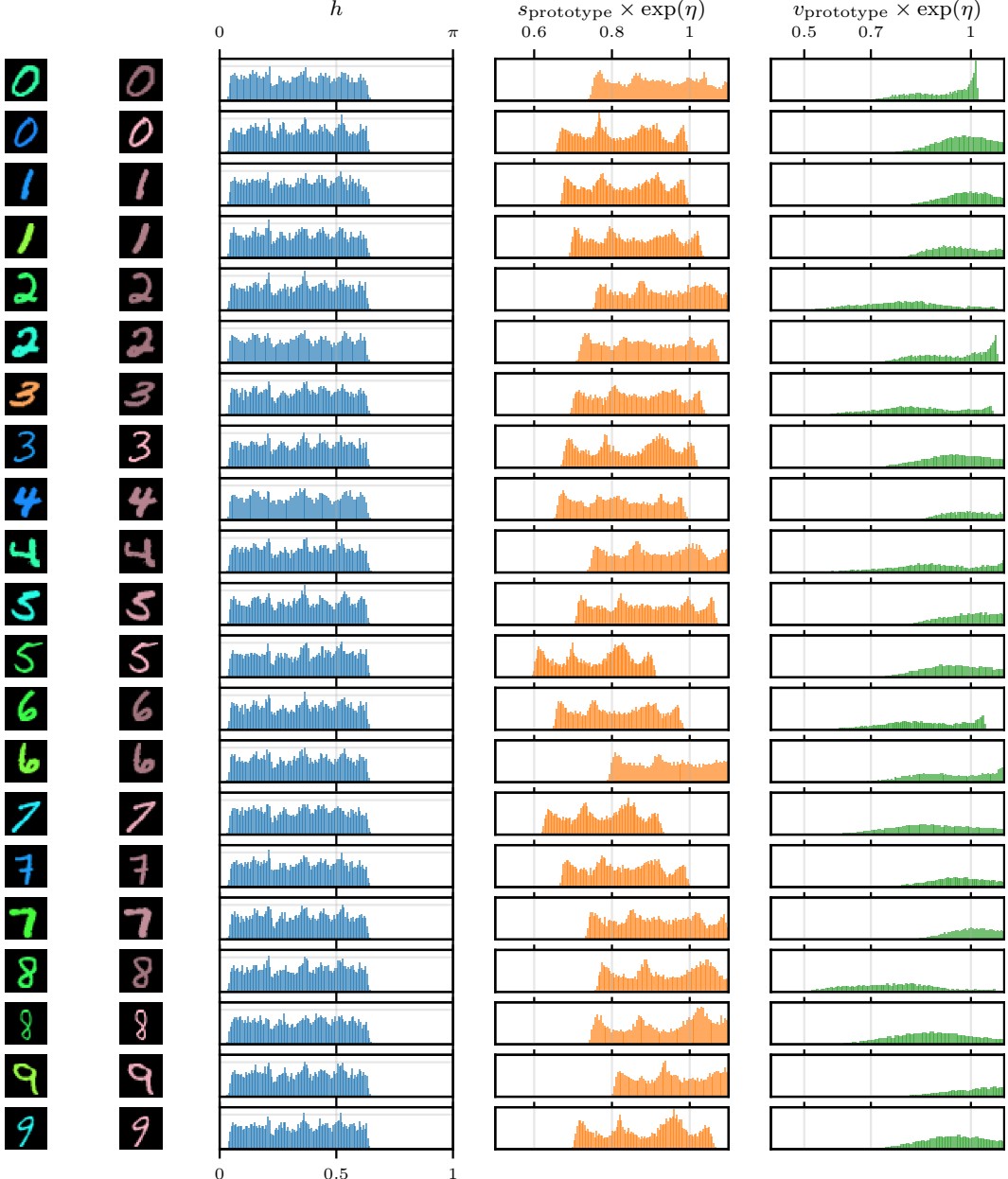

Figure 22: From left to right, test examples from MNIST with added hue in the range 0 to $0.6\pi$, and saturation scaled by a factor in 0.6 to 0.9, their prototypes, and the corresponding marginal distributions over hue, saturation, and value.

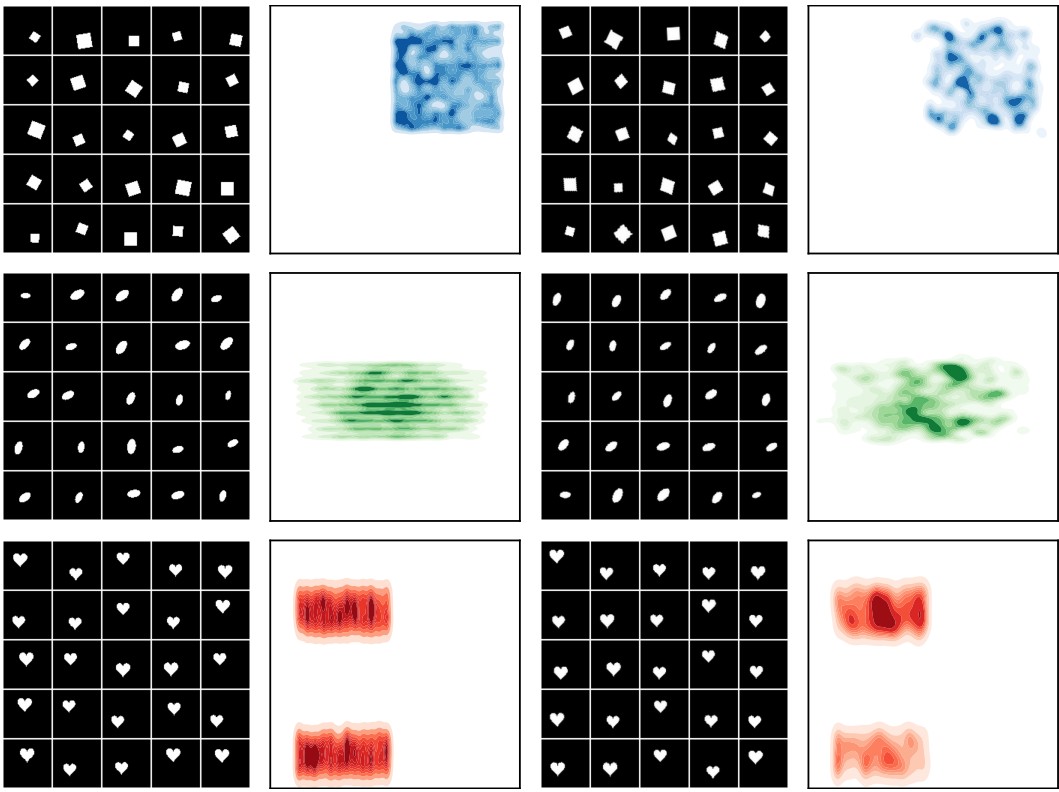

Figure 23: From left to right, samples from dSprites, the empirical distribution over the positions of the sprites, sprites resampled using our SGM, and the empirical distributions over the resampled sprites' positions. We see that the resampled sprites are visually very similar to the original sprites in terms of sizes, rotations, and positions. Furthermore, we see that the empirical distributions match in terms of ranges, although they are imperfect in density.

