# OpenReview forum: "A Generative Model of Symmetry Transformations"
_NeurIPS.cc/2024/Conference — NeurIPS 2024 poster_

### Official Review · Reviewer_GGon · 2024-07-02

**Soundness:** 3
**Presentation:** 3
**Contribution:** 3
**Rating:** 6
**Confidence:** 5

**Summary:**

The paper aims to model the data distribution along each group action orbit. The proposed two-stage method first uses a self-supervised loss to learn an invariant function that maps each sample to its prototype and then uses a normalizing flow to learn the distribution along each orbit (i.e. conditional distribution on each prototype). To acquire the full data distribution, one trains a generative model on the prototypes and composes it with the conditional model. The method shows merits in terms of modeling the prototype-dependent distributions of symmetry transformations in the experiments on small-scale datasets.

**Strengths:**

* The paper presents a novel framework for learning the distribution of symmetry transformations. Using a flow generative model allows for much more expressivity, compared to the parametrized Gaussian or uniform distributions considered in [1].
* The proposed method is easy to understand. The design choices are well-motivated and clearly described.
* Experiments show interpretable results supporting the claims that
    * The proposed training objective can lead to approximately invariant prototypes.
    * The orbital distributions may vary for different prototypes.
    * The proposed method can increase the data efficiency for datasets with certain degrees of symmetry transformations.

**Weaknesses:**

* My main concern is the practical applications of the method. Currently, the experiments are done on small image datasets, e.g. dSprites and MNIST. Can the authors identify some more complicated tasks where modeling the symmetry transformations could be beneficial?
* Also, the dimensionality of symmetry transformations is generally much smaller than the dimensionality of the data manifold. For example, an image dataset may have a 28*28 pixel space and a lot of possible variations in there, while the symmetry of planar affine transformations only accounts for 6 dimensions of variations. Generative modeling is difficult because of the high dimensionality. I'm unsure if it's worth the effort to use a generative model to learn the low-dimensional distribution on a group orbit.
* (This may be just my personal preference) I find the rotated and colored characters throughout the text a bit distracting. The normal texts have already made things pretty clear. Those characters may be great for intuition but are less accurate and formal.
* Currently, the parameterizations for different symmetry transformations seem ad-hoc. E.g. affine transformations are represented in the affine matrix. However, symmetry groups have different structures (which can, for example, be reflected by the structure constants of the Lie algebra) and require different ways of parametrizing distributions. The authors should address this aspect and possibly discuss some related works, e.g. [2].

**Questions:**

* Regarding the experiment setting, currently the MNIST dataset is manually transformed. I'd expect smaller variations in the original dataset. In that case, would the proposed method result in less performance increase?

## References

[1] Benton, Gregory, et al. "Learning invariances in neural networks from training data." Advances in neural information processing systems 33 (2020): 17605-17616.

[2] Falorsi, Luca, et al. "Reparameterizing distributions on lie groups." The 22nd International Conference on Artificial Intelligence and Statistics. PMLR, 2019.

**Limitations:**

The authors have discussed the limitations.

---

> ### Author Rebuttal · Authors · 2024-08-07
>
> > My main concern is the practical applications of the method. Currently, the experiments are done on small image datasets, e.g. dSprites and MNIST. Can the authors identify some more complicated tasks where modeling the symmetry transformations could be beneficial?
>
> Please see our general response, in which we provide detailed motivation as to why our experiments are sufficiently interesting and informative. In short, we have already gone beyond many published related works, our choices of datasets cover a range of dataset sizes and data dimensionalities, and the GalaxyMNIST dataset contains high-dimensional natural images while providing challenges due to its small number of observations and large number of symmetries.
>
> > Also, the dimensionality of symmetry transformations is generally much smaller than the dimensionality of the data manifold. For example, an image dataset may have a 28*28 pixel space and a lot of possible variations in there, while the symmetry of planar affine transformations only accounts for 6 dimensions of variations. Generative modeling is difficult because of the high dimensionality. I'm unsure if it's worth the effort to use a generative model to learn the low-dimensional distribution on a group orbit.
>
> We agree that the data manifold is larger than the dimensionality of the transformations and that one of the main challenges of generative modeling is high dimensionality. This is actually one of the main motivations for this work. Our hypothesis is that decomposing the generative modeling task into an “easier” symmetry modeling task and a more complicated task of modeling all of the other sources of variation, will provide benefits such as data-efficiency and improved model fit. Figures 11 and 12 confirm this hypothesis, with our symmetry augmented VAEs easily outperforming vanilla VAEs without this inductive bias. In addition to these improvements, this decomposition provides benefits in interpretability of the latent code, and the ability to easily generate realistic “data-augmentations” of any observation. Whether or not it is worth the effort depends on the particular application at hand, however, we are confident that the strengths of our method could be useful in several settings. Finally, we hope that our “novel framework” and “easy to understand” method will result in further interesting developments within the field.
>
> > (This may be just my personal preference) I find the rotated and colored characters throughout the text a bit distracting. The normal texts have already made things pretty clear. Those characters may be great for intuition but are less accurate and formal.
>
> We appreciate your input on this, and understand that their inclusion might not be to everyone’s tastes. However, prior to their inclusion, we received several pieces of feedback that the corresponding sections were hard to understand. Since their inclusion we have found that readers are much more easily able to understand the text. Furthermore, yourself as well as reviewers d330 and rf86 have noted the clarity of our text. Nonetheless, we will take your input to heart, and for the camera-ready version we will carefully consider the inclusion of each of these characters.
>
> > Currently, the parameterizations for different symmetry transformations seem ad-hoc. E.g. affine transformations are represented in the affine matrix. However, symmetry groups have different structures (which can, for example, be reflected by the structure constants of the Lie algebra) and require different ways of parametrizing distributions. The authors should address this aspect and possibly discuss some related works, e.g. [2].
>
> We will happily include a discussion of [2] (and any other related works you suggest). However, we note that we use both affine matrices and the corresponding Lie algebra constants when representing affine transformations, as each representation has pros and cons and the two representations can easily be interchanged. For instance, our inference and generative networks output the Lie algebra constants, since these are low dimensional and easy to constrain (if necessary). On the other hand, we use affine matrices for transformation composition. Both of these choices are discussed in Section 3.1. Similarly, our representation of color transformations is chosen to make composition of transformations simple. See Appendix D.6 for further details.
>
> > Regarding the experiment setting, currently the MNIST dataset is manually transformed. I'd expect smaller variations in the original dataset. In that case, would the proposed method result in less performance increase?
>
> Your intuition is correct – the smaller the degree of transformation present in the dataset, the less our method is expected to improve data efficiency. This can be observed in Figure 11 – we see that as more rotation is added to the dataset, the performance gap between AugVAE and VAE becomes larger. However, our results of GalaxyMNIST (which don’t include any manual transformations) demonstrate that our method still performs well in natural settings. Finally, we note that the performance gain from our method can be increased by including a wider range of transformations (e.g., learn both color and affine transformations together).

---

> > ### Comment · Reviewer_GGon · 2024-08-07
> >
> > Thank you for your detailed response. Most of my concerns have been addressed.
> >
> > Regarding your second point, I agree that the proposed method can improve sample efficiency and model fit by decoupling a small number of variations described by symmetry. I just feel that since the orbital distributions are low-dimensional, using a complex generative model may not be the most simple and efficient way to achieve this. This is a somewhat subjective matter, and I will not make further comments.
> >
> > Overall, I maintain my opinion that this is a well-written paper with clear motivation and reasonable experiment results. I will keep my current recommendation.

---

> > > ### Author Response · Authors · 2024-08-12
> > >
> > > > Regarding your second point, I agree that the proposed method can improve sample efficiency and model fit by decoupling a small number of variations described by symmetry. I just feel that since the orbital distributions are low-dimensional, using a complex generative model may not be the most simple and efficient way to achieve this. This is a somewhat subjective matter, and I will not make further comments.
> > >
> > > While these distributions are low-dimensional, they can often be complex, with dependencies between the different dimensions. For instance, in the case of affine transformations, there is a non-trivial relationship between rotation and shifting/scaling. As a result, we found that using a coupled neural spline flow was actually required for accurately modeling the distribution over transformation parameters. In cases where the dimensions do not interact it is possible to greatly simplify the model (e.g., a flow with a single layer and no coupling). In other words, we view the complexity of the models for transformation parameter distributions as a problem-specific design choice rather than an inherent feature of our method.

---

### Official Review · Reviewer_d33o · 2024-07-04

**Soundness:** 3
**Presentation:** 3
**Contribution:** 2
**Rating:** 4
**Confidence:** 5

**Summary:**

This paper proposes a Symmetry-aware Generative Model (SGM), aiming to learn (approximate) symmetry presented in a data. The model achieves this by mapping each sample onto a prototype—a unique representative on the group orbit—and learning the conditional distribution over its group orbit through maximum likelihood estimation.

**Strengths:**

The paper is easy to follow

**Weaknesses:**

1. Initially, I thought the paper aimed to address data-efficient learning of distributions with unknown approximate group symmetry. However, the scope of the paper is quite limited. Essentially, it only aims to learn the conditional distribution of symmetry transformations over the group orbits. In other words, the paper's primary focus is on augmenting training samples over their group orbits according to this supposedly accurate conditional distribution.

2. To demonstrate that the model is functioning as intended, the paper should provide results showing whether this conditional distribution is learned correctly. Unfortunately, this is not shown in any of the examples. The "realistic-looking" generated samples depicted in the figures are merely group-transformed versions of the given samples, which is why they appear realistic.

3. One of the main components of the paper is projecting each sample onto a unique representative of its group orbit, achieved through a so-called transformation inference function \( f_w: X \to H \), where \( H \) is the (potentially large) group. The paper trains this function \( f_w \) in a self-supervised manner through equation (6) to produce unique prototypes. However, this approach is completely unnecessary, as any equivariant function \( f_w: X \to H \) should already accomplish this.

4. The authors claim that one advantage of their approach is handling data sets with unknown symmetries. However, the paper only deals with (2D) rotation and scaling, as well as color transformations, which limits its generalizability.

5. Additionally, the images used in the paper, such as MNIST, are always compactly supported and vanish at the boundary. This makes data augmentation using rotation and scaling feasible. It is unclear how the proposed method would fare when dealing with realistic images that exhibit boundary effects after symmetry transformations.

6. The combination of the proposed model (which supposedly "learns" how to augment data) with a VAE is also unconvincing. While it might be better than directly applying data augmentation (as small images can become even smaller), without demonstrating that the conditional distribution is learned accurately, this combined model could very well learn an incorrectly symmetrized distribution.

**Questions:**

Please see the above section

**Limitations:**

Yes

---

> ### Author Rebuttal · Authors · 2024-08-07
>
> (1) Learning this conditional distribution is non-trivial. We show that a GAN-based method fails (see Appendix F.1). Furthermore, in sections 3.1 and 3.2 we discuss design and implementation pitfalls that make this challenging. We provide guidelines as to how to resolve them. We consider these sections to be a core part of our contributions.
>
> Our method has several use cases, of which we focus on two:
> * Learning distributions over transformations.
> * Leveraging those transformations to improve deep generative models.
>
> Our experiments demonstrate success in both 1 and 2.
>
> (2) For evidence that our model is functioning correctly see section 4.1 and Appendix F.2. For MNIST, we don’t know the ground truth distribution over the transformations (though our method provides sensible results). However, figures 19 and 20 in Appendix F.2, show results for colored MNIST and dSprites, where we control the transformations. E.g., for colored MNIST, our model learns uniform distributions of hue transformations in exactly the range we added to the dataset. The same is true for dSprites.
>
> (3) Yes, any equivariant function could work, but in general it isn’t obvious how to construct such a function (e.g., a function with equivariance from images to transformation parameterizations for HSV/general affine transformations). Thus, we provide a *general* method for learning such equivariances. In “Invariance of f_ω and the prototypes” we mention the possibility of using an architecture with a subset of equivariances directly built in. We will further clarify this in the camera-ready version.
>
> (4) While we used affine and color transformations in our experiments, our method is general to any transformation for which (approximate) composition and inverse operations exist. This is a very general class of transformations.
>
> We have shown that our method can successfully learn *3* affine transformations (rotation, shift, scale) and *3* color transformations (hue, saturation, value). These transformations have very different properties. Furthermore, all of these transformations can be learnt concurrently (for a total of 8 transformation parameters).
>
> This goes beyond several related works which often focus only on affine transformations, and when color transformations are considered it is usually in isolation. E.g.:
> * “Disentangling images with Lie group transformations and sparse coding” by Chau et al.  (from Reviewer TzHS), van der Ouderaa and van der Wilk [2022], Immer et al. [2022] – affine only,
> * Benton et al. [2020], Keller and Welling [2021] – affine and color separately.
>
> (5) Boundary effects may be a problem, by providing a trivial solution for the inference net, allowing it to infer the transformation parameters by ignoring the contents of the image and instead focusing only on the boundaries.
>
> Our results for GalaxyMNIST–where images do not vanish at the boundaries due to other galaxies, stars, and background noise–demonstrate that these boundary effects do not *necessarily* pose a problem for our model as shown in Figures 8d and 12.
>
> Note, there are many transformations (e.g., hue, saturation, and value) for which this isn’t an issue.
>
> Nonetheless, we acknowledge this potential limitation, and we'll include a discussion in the camera-ready. This is also a limitation of some existing methods (e.g., LieGAN [Yang et al., 2023], where the edge effects could be used by the discriminator to easily distinguish between real and generated data). It wasn’t the goal of our work to address this limitation.
>
> (6) In addition to the evidence provided in our response to question 2, the fact that our SGM-VAE hybrid models outperform a standard VAE baseline (measured by *test-set* IWLB) shows that the model has learnt a better distribution over transformations than the vanilla VAE. If an incorrect distribution were learned, these models would place too much probability mass on uncommon transformations (and vice versa) which would negatively impact the marginal likelihood of the test data.
>
> > Rating: 2: Strong Reject: For instance, a paper with major technical flaws, and/or poor evaluation, limited impact, poor reproducibility and mostly unaddressed ethical considerations.
>
> We are surprised that we received this score of 2, which we do not believe reflects our work's quality and contributions. This is especially surprising, given the other reviewers' positive scores and comments, which contradict this score. We hope you will reconsider this score. If you are still convinced that the score is accurate, we kindly ask that you address the following:
> * List the flaws in the paper that you consider major,
> * Explain why you think our evaluation is poor (in contrast with reviewer rf86 who said our paper has “thorough experimental validation on a wide range of datasets” and reviewer GGon who said our paper’s “experiments show interpretable results supporting the claims”), by providing examples of similar papers who’s evaluation we should aim to emulate,
> * Explain why the impact of the paper will be limited (in contrast with reviewers GGon and rf86 who noted the novelty of our work), by citing works that detract from our novelty,
> * Let us know what we can do to improve our reproducibility, given that we provide code and an extensive appendix explaining our experimental setup (“extensive details on the experimental set-ups making them highly reproducible” according to reviewer rf86), or
> * Point to any ethical issues in our paper.
>
> The review also lists “easy to follow” as the only strength of the paper, which we found surprising given the strengths noted by all other reviewers and the contributions the paper makes upon prior work, which are:
> * A novel generative model (SGM) of the symmetry transformations,
> * A learning algorithm for our SGM,
> * The intuition behind and practical tips for our SGM, and
> * The extensive experimental results that show our model can accurately learn prototypes and distributions over transformations.

---

> > ### Comment · Reviewer_d33o · 2024-08-08
> >
> > I thank the authors for their detailed response. After reviewing the paper and the feedback, I acknowledge that my previous review had flaws. Specifically, Figures 14, 19, and 20 in the appendix convincingly demonstrate that the proposed method learns the correct conditional distribution, and these figures slipped through my first review. Since my primary concern was the lack of clear evidence that the method achieves this rather than merely **augmenting** training data and **displaying** visually realistic samples, I find it necessary to raise the score.
> >
> > However, I still have several concerns:
> >
> > 1. While the authors provide a general method for learning equivariant $f_w$, it would be beneficial to compare it with a hardwired equivariant $f_w$ to show its competitive performance.
> >
> > 2. I remain unconvinced that boundary effects are insignificant. The method’s core idea, when used as a preprocessing step for VAE, is to augment samples in the group orbit accurately. Although GalaxyMNIST does not vanish at the boundary, it primarily consists of sparse objects on a black background.
> >
> > 3. The paper assumes prior knowledge of the group symmetry in the dataset. If I understand correctly, the comparison to [Yang et al. 2023] might be unfair, as [Yang et al. 2023] genuinely learns the general linear group, whereas SGM assumes the transformation group is limited to the rotation and scaling subgroup.
> >
> > Nonetheless, since my main concern about demonstrating the correct conditional distribution learning (points 1, 2, and 6 in my original review) has been addressed, I will substantially raise the score while still reserving my opinion on other aspects.

---

> ### Author Response · Authors · 2024-08-12
>
> > While the authors provide a general method for learning equivariant, it would be beneficial to compare it with a hardwired equivariant to show its competitive performance.
>
> We want to stress that we make no claims that our method for learning an equivariant f_w will match the performance of a hardwired f_w, in fact, we acknowledge in the paper that having some equivariances hardwired would likely lead to an increase in performance.
>
> However, we agree that comparison with a hardwired equivariant $f_\omega$ would be interesting. If you have suggestions or know of relevant work for how to parameterise such an $f_\omega$, we'd be happy to run those experiments. This would have been easy if $f_\omega$ was a function from image to image space equivariant to translations/rotations, but we don't know of any work that does this for functions from an image space to transformation-parameter space.
>
> > I remain unconvinced that boundary effects are insignificant. The method’s core idea, when used as a preprocessing step for VAE, is to augment samples in the group orbit accurately. Although GalaxyMNIST does not vanish at the boundary, it primarily consists of sparse objects on a black background.
>
> As we mentioned in our previous response, we acknowledge that this is a potential limitation of our method, but note that this is also a limitation of existing published methods. Thus, we hope that we will not be held to a higher standard for publication.
>
> > The paper assumes prior knowledge of the group symmetry in the dataset. If I understand correctly, the comparison to [Yang et al. 2023] might be unfair, as [Yang et al. 2023] genuinely learns the general linear group, whereas SGM assumes the transformation group is limited to the rotation and scaling subgroup.
>
> To clarify, in our experiments for this paper our SGM covers rotation, scaling, *and shifting*. This makes it slightly less flexible than the LieGAN of Yang et al. [2023], since their method is also able to learn shearing and flipping. However, we feel that the comparison is largely fair, since we are not reporting any quantitative results, and instead focus on qualitative comparisons. From these qualitative comparisons, it is clear that fundamental differences between our two approaches (e.g., our SGM learning conditional distributions) are the dominant reason for different behavior, rather than the choice of assumed transformation group. We also note that our SGM is also capable of learning shearing and flipping, we simply didn't include these transformations in our experimental results. However, expending our assumed transformation group to include these is trivial.
>
> > Nonetheless, since my main concern about demonstrating the correct conditional distribution learning (points 1, 2, and 6 in my original review) has been addressed, I will substantially raise the score while still reserving my opinion on other aspects.
>
> Thank you for increasing your score and engaging with the rebuttal. We very much appreciate the discussion and we are glad that the rebuttal has already addressed many of your concerns. We hope that we have addressed your remaining concerns and that you will consider increasing it further.

---

### Official Review · Reviewer_TzHS · 2024-07-06

**Soundness:** 3
**Presentation:** 3
**Contribution:** 2
**Rating:** 6
**Confidence:** 4

**Summary:**

The paper proposes a generative model that disentangles the latent space into a group-invariant part -- the latent for the prototype -- and another part which represent a group element that can be applied to the prototype to reconstruct the input. A key novelty is to simultaneously learn to predict a distribution for each input over the group elements based on data. This input-dependent distribution, in principle, can be then used to generate new data that better aligns with the true group distribution.

**Strengths:**

The overall architecture seems interesting and sound and the idea of learning probability distribution over the group elements is especially interesting. The results show that the distributions are dependent on the input and are meaningful for the datasets used in the experiments.

**Weaknesses:**

Some parts of the paper are confusing to me.

What is the architecture for the part that predicts the distribution over the group elements? I saw one or two mentions of normalizing flows, but that is not enough to understand the details. The figure is quite unclear about it. Shouldn't the input image be also an input to the network that predicts $p_\psi(\eta|x)$? Also, the paper does not seem to have any information on the loss function used to train the network that predicts $p_\psi(\eta|x)$. A lot more clarity is needed for understanding these important details.

I feel that the exact new contributions of the paper are a little unclear given many previous works doing similar things. Many of them are also mentioned by the authors in the appendix, but the contrasts with this paper are not clear. My understanding is that it is a generative model and can find the right distribution over the group elements directly from data, but there are earlier works also looking into these aspects. Some discussion here would be useful.

Xu et al., Group Equivariant Subsampling
Romero and Lohit, Learning Equivariances and Partial Equivariances from Data
Shu et al., ,Deforming autoencoders: Unsupervised disentangling of shape and appearance
Chau et al., Disentangling images with Lie group transformations and sparse coding

Even earlier work by Grenander, Mumford and others on Pattern Theory discusses generative models and probability distributions over groups, but these are not based on neural networks.

I think there should be at least one dataset which is a little more challenging like an image recognition dataset of more natural images. This can also help in understanding the limitations of the method. Another experiment showing the usefulness of the generative model being able to generate samples that respect the data distribution is also going to be useful.

**Questions:**

Please address the weaknesses I have listed above.

**Limitations:**

I don't think the paper mentions the limitations explicitly, which the authors should try to.

---

> ### Author Rebuttal · Authors · 2024-08-07
>
> > What is the architecture for the part that predicts the distribution over the group elements?
>
> In short, we use an MLP with hidden layers of dimension [1024, 512, 512] as a shared feature extractor. These shared features are fed into another MLP hidden layers of dimension [256, 256] that outputs a mean and a standard deviation. The shared features are also used by MLPs with a single hidden dimension of size 256 that output the flow parameters at each layer of the neural spline flow, which has 6 bins in the range [-3, 3].
>
> Please see Appendix D for further details about the exact NN architectures used for both the inference and generative networks. If you would like, we can include some of the details in the main text (given the extra page allowed for the camera-ready version). If so, please let us know what you would find most useful.
>
> > Shouldn't the input image be also an input to the network that predicts
>
> We are not sure which network you are referring to. The inference network does indeed take the original image as an input. On the other hand, the generative network, which represents the distribution of $p(\eta|\hat{x})$, must take $\hat{x}$, rather than $x$ as input. One way to think of why it must take $\hat{x}$, rather than $x$, is that we want to learn the distribution of transformations present in the whole dataset. This distribution should not change depending on the transformation applied to an individual input (e.g., the rotation of a specific digit) but rather an aggregate of transformations applied to all the digits of the same type. Since the distribution is capturing a “dataset-level” transformation, it should depend on a representation of data that is invariant to those transformations (i.e., the prototype $\hat{x}$).
>
> > Also, the paper does not seem to have any information on the loss function used to train the network that predicts
>
> We are not sure which network you are referring to. In section 2.1, under “Transformation inference function” we discuss how the inference network  is trained. In short, we use a SSL loss depicted in Figure 4 (and equation 6). Similarly, under “Generative model of transformations” we discuss how the generative network is trained. In short, we use the inference network to generate data and we then fit the conditional flow with maximum likelihood. Both loss functions and training methods are summarized in Algorithm 1.
>
> > A lot more clarity is needed for understanding these important details.
>
> Please let us know if there are any other clarifications required.
>
> > I feel that the exact new contributions of the paper are a little unclear...
>
> Thank you for this feedback. We will update the related work sections to clarify this better. We provide a short summary below.
> While the goals of our paper–namely, (1) unsupervised learning a distribution over arbitrary symmetry transformations present in a dataset, and (2) leveraging this distribution for improved data-efficiency in deep generative models–and the techniques we employ to do so–(A) our SSL objective for learning invariant representations and (B) maximum likelihood learning of flexible flow models for the distributions over *partial*-symmetries–are similar/related to a wide range of existing work, to the best of our knowledge ours is the only work that incorporates all of 1, 2, A, and B.
>
> Related methods tend to not learn the distribution of interest (e.g., [Winter et al., 2022] and “Deforming autoencoders”), do not learn invariant prototypes (e.g., [Yang et al., 2023]), focus on the discriminative rather than generative setting learning setting (see below), or construct deep-generative models for specific symmetries (e.g., [Kuzina et al., 2022] and [Vadgama et al., 2022]).
>
> > Xu et al., Group Equivariant Subsampling Romero and Lohit, Learning Equivariances and Partial Equivariances from Data Shu et al., ,Deforming autoencoders: Unsupervised disentangling of shape and appearance Chau et al., Disentangling images with Lie group transformations and sparse coding
>
> Thank you for providing these additional items of related work. We will happily include them in our camera-ready manuscript. We provide brief discussions for each paper in the global comment above.
>
> > Even earlier work by Grenander, Mumford and others on Pattern Theory discusses generative models and probability distributions over groups, but these are not based on neural networks.
>
> If you provide specific missing references, we’d be more than happy to include them in our discussion
>
> > ... at least one dataset which is a little more challenging like an image recognition dataset of more natural images... experiment showing the generative model being able to generate samples that respect the data distribution...
>
> Please see our general response, in which we provide detailed motivation as to why our experiments are sufficiently interesting and informative. In short, we have already gone beyond many published related works, our choices of datasets cover a range of dataset sizes and data dimensionalities, and the GalaxyMNIST dataset contains high-dimensional natural images while providing challenges due to its small number of observations and large number of symmetries.
>
> Please see Appendix F.2 for additional experiments showing that the generated samples respect the data distribution.
>
> > I don't think the paper mentions the limitations explicitly...
>
> We have provided several limitations, e.g., in footnote 1 we discuss how our generative model might not always match the true generative process, and in our conclusion we discuss our need to pre-specify a super-set of possible symmetries.
>
> In the camera ready version, we will also include a discussion of potential limitations of our method due to boundary effects (see our discussion with reviewer d33o).
>
> > Rating: 5: Borderline accept
>
> Please let us know if we have successfully addressed your concerns. If so, we would appreciate it if you would consider increasing your score.

---

> > ### Author Response · Authors · 2024-08-09
> >
> > We thank the reviewer again for the effort they put into reviewing our paper. Since there are only a few working days left for the discussion period, we would like to ask if our response satisfied the reviewer's concerns. If that is the case, we kindly invite them to raise their score. If there are still any remaining concerns, we are happy to discuss them here.

---

> > ### Comment · Reviewer_TzHS · 2024-08-13
> > **Thank you for the response**
> >
> > The responses are indeed helpful in understanding some parts of the paper that I didn't before.
> >
> > I understand that the architecture details and other training details are in the appendix. Yes, it would be helpful if some of the important details are moved to the main paper.
> >
> > I understand the connection to existing work better.
> >
> > I still think having one dataset with more challenging images is important. At the very least, it shows what the limitations of the method are. Even a dataset like CIFAR-100 with 32 x 32 color images or CUB-200-2011 or PatchCamelyon would be interesting to show some experimental results on.
> >
> > A separate paragraph on limitations would be good to have rather than a footnote. For example, the authors say that running their method on more challenging datasets is computationally difficult. I don't understand if this is because the method is inherently more complex because of self-supervised learning objectives etc. or some computational constraints the authors may face.
> >
> > This is not too important, but this is one reference for Pattern Theory: Pattern Theory, the Stochastic Analysis of Real World Signals by Mumford and Desolneux. But the authors should not feel compelled to include this is in the paper if they don't believe it to be very related.
> >
> > Overall, I think the reviewers have addressed my concerns to some extent. I think it is important to have some experimental validation of more challenging datasets. Based on the above, I will raise my score to a weak accept.

---

> ### Author Response · Authors · 2024-08-13
>
> Thank you for your engagement with the review process and for increasing your score as a result – we are very grateful!
>
> We'll make sure to incorporate your feedback. Specifically, to
> * include some more of the architecture and training details in the final manuscript,
> * try our best to run some additional experiments with the datasets you have suggested (PatchCamelyon looks particularly interesting),
> * include your suggested reference (Mumford and Desolneux), and
> * include a dedicated paragraph on the limitations we have discussed in our previous response.

---

### Official Review · Reviewer_rf86 · 2024-07-12

**Soundness:** 3
**Presentation:** 4
**Contribution:** 4
**Rating:** 8
**Confidence:** 4

**Summary:**

The paper proposes a generative model of symmetry transformations. The work leverages recent parameterizations based on group theory to define a generative model, in which relaxed symmetry becomes a latent variable over which inference can be performed.

**Strengths:**

The paper is very well written and proposes an elegant method that leverages group theoretical framework to mathematically define approximate equivariances, combined with a probabilistic approach to guide discovery of symmetry transformations and learn a prototype.

Lastly, the reviewer notes that the paper is very well-written and has beautiful illustrations which guide the intuitive understanding of the generative model and the concept around learning a prototype.

Practicality
It seems that the proposed SSL objective has computational benefits (easy to scale) as well as benefits in performance (ELBO objective only worked for rotations). Is this true, or am I reading this too positively? What would be potential disadvantage of choosing such loss?

Experiments
The paper provides very thorough experimental validation on a wide range of datasets. The appendices provide extensive details on the experimental set-ups making them highly reproducible.

**Weaknesses:**

Intuition behind the objective.
Experimentally the paper demonstrates that the proposed SSL objective is very effective. Apart from the computational benefits of such approach, it also seems to improve overall performance (optimizing ELBO only worked for rotations). To me it is not entirely clear why this would be the case, and it would be interesting to provide some more explanations on this - if known, of course. The first appendix was very helpful in providing context in relation to directly optimizing the ELBO.

Types of transformations.
The paper does not provide a lot of discussion on how the density of transformations is parameterized. In case of rotations, how is the normalizing flow constrained to remain smooth in the Lie algebra? Is the approach mostly targeted to simple (e.g. affine) groups?

**Questions:**

1. The connection in App. A. offers a nice connection to the objectives used in some of the prior work. It references earlier approaches by the authors [Authors, 20XX] in which the ELBO is directly optimized and hypothesizes that ‘ averaging of many latent codes makes it difficult to learn an invariant representation a without throwing away all the information in x’ . Could the authors elaborate a bit more on this hypothesis / is this backed by any experiment?
2. Number of samples. The x-mse in number of samples is optimal for 5 samples. Don’t we expect this table to be monotonically decreasing in number of samples?
3. Overfitting on p(\eta | x). For very flexible distributions of transformations, isn’t there a risk of overfitting on the parameterization? Please correct me if I have missed something in the method which counteracts this.

**Limitations:**

The paper offers a strong contribution proposing a probabilistic generative model that describes that as coming from transformed latent prototypes.

---

> ### Author Rebuttal · Authors · 2024-08-07
>
> > Practicality It seems that the proposed SSL objective has computational benefits... Is this true, or am I reading this too positively? What would be potential disadvantage of choosing such loss?
>
> Your understanding is correct. We provide additional discussion below. Regarding disadvantages, the SSL objective does have a few pathologies and potential ‘gotchas’ that are not present in the ELBO setting. E.g., see “Partial invertibility” in Section 3.1 and Appendix B.
>
> > The connection in App. A. offers a nice connection to the objectives used in some of the prior work. It references earlier approaches by the authors [Authors, 20XX] in which the ELBO is directly optimized and hypothesizes that ‘ averaging of many latent codes makes it difficult to learn an invariant representation a without throwing away all the information in x’ . Could the authors elaborate a bit more on this hypothesis / is this backed by any experiment?
>
> In trying to scale that method, we observed that as we added additional transformations or increased the range of those transformations (e.g., increasing rotation from 0-pi to 0-2pi) that performance (as measured by the ELBO and reconstruction loss) degraded to the point that when using the 5 transformations applied to MNIST in this paper, the model was unable to reconstruct the digits at all. Instead it became stuck in a local optima in which the ‘reconstructions’ were all circles and rings of various sizes depending on the input image. (We’ll include some of these figures in the camera-ready version.)
>
> In other words, the averaged latent code was successfully throwing away (e.g.,) rotation information but was also throwing away all of the information that actually identified each digit. This led us to the hypothesis that averaging of latent codes makes it difficult to learn representations that only throw away the symmetry data.
>
> Our current method is directly motivated by this observation – we aimed to develop an algorithm that could produce invariant representations without latent-space averaging. The success of our SSL objective is indirect evidence to support our hypothesis.
> That said, our SSL algorithm has another advantage over ELBO learning – it decouples learning an invariant representation of x from reconstruction of x. That is, for ELBO learning to successfully learn an inference network, one ultimately needs a good generative network (and vice-versa). However, learning a network to generate examples given latent codes is a challenging inverse learning task. This is an observation that was also made by Dubois et al. [2021], who found that an SSL based objective was superior to an ELBO based method for learning invariant representations in the context of compression.
>
> > Number of samples. The x-mse in number of samples is optimal for 5 samples. Don’t we expect this table to be monotonically decreasing in number of samples?
>
> Thanks for the question, we will clarify this in the camera-ready version. The table is not likely to be *monotonically* decreasing, due to the fact that there is random noise in each training run (i.e., due random NN initialization, etc.). That said, we would expect that it will decrease on average as the number of samples is increased. We chose 5 samples not because it provides the lowest loss, but rather because it provided a good trade-off between lower loss and increased compute cost.
>
> > Overfitting on $p(\eta | x)$. For very flexible distributions of transformations, isn’t there a risk of overfitting on the parameterization? Please correct me if I have missed something in the method which counteracts this.
>
> In practice, for the inference network $p(\eta | x)$ we found that there were no issues with overfitting (the more expressive the network and the longer we trained, the better we found the test-set performance became). This is likely due to two things: (1) our SSL loss, which has ‘baked-in data-augmentation’ in the form of random transformations applied to x, and (2) learning a function with equivariance to arbitrary transformations is hard.
>
> On the other hand, for the generative network $p(\eta | \hat{x})$, we did observe overfitting. We addressed this by using a validation set to optimize several relevant hyper-parameters (e.g., dropout rates, number of flow layers, number of training epochs, etc.).
>
>
> > Rating: 8: Strong Accept: Technically strong paper, with novel ideas, excellent impact on at least one area, or high-to-excellent impact on multiple areas, with excellent evaluation, resources, and reproducibility, and no unaddressed ethical considerations.
>
> Please let us know if you have any remaining concerns.

---

> > ### Author Response · Authors · 2024-08-09
> >
> > We thank the reviewer again for the effort they put into reviewing our paper. Since there are only a few working days left for the discussion period, we would like to ask if our response satisfied the reviewer's concerns. If that is the case, we kindly invite them to raise their score. If there are still any remaining concerns, we are happy to discuss them here.

---

> ### Comment · Reviewer_rf86 · 2024-08-12
>
> We thank the authors for the further clarifications and answering my answers and hope these are included as discussions the final manuscript. I regard this a technically strong paper, with novel ideas and a good execution, and therefore keep my recommendation for acceptance with score rating 8.

---

> > ### Author Response · Authors · 2024-08-13
> >
> > Thank you for your strong endorsement of our paper. We will be sure to include all of the answers to your questions as additional discussion on the final camera-ready manuscript.

---

### Author Rebuttal · Authors · 2024-08-07

We thank the reviewers for their time and constructive feedback on our paper. We are pleased that the reviewers have highlighted the quality of our writing (rf86, d33o, GGon), experimental evaluation (rf86, TxHS, GGon), the novelty, elegance and interestingness of our work (rf86, TxHS, GGon). We would like to highlight the following quotes from the reviews:
* “The paper is very well written and proposes an elegant method”, “The paper is very well-written and has beautiful illustrations”, “The paper provides very thorough experimental validation on a wide range of datasets”, “The appendices provide extensive details on the experimental set-ups making them highly reproducible” – reviewer rf86
* “The overall architecture seems interesting and sound”, “the idea of learning probability distribution over the group elements is especially interesting” – reviewer TzHS
* “The paper is easy to follow” – reviewer d33o
* “The paper presents a novel framework for learning the distribution of symmetry transformations.”, “The proposed method is easy to understand.”, “The design choices are well-motivated and clearly described.”, “Experiments show interpretable results supporting the claims …” – reviewer GGon

We also acknowledge a common criticism among most of the reviewers (TzHS, d33o, GGon) that the paper would be improved with the addition of a larger-scale dataset with ‘natural’ images. We believe our experiments are sufficiently interesting and informative for the following reasons.

Many *published* related works do not go beyond image datasets of similar size/dimensionality to MNIST and dSprites. Examples of this include:
* “Disentangling images with Lie group transformations and sparse coding” by Chau et al., and “Group Equivariant Subsampling” by Xu et al. (mentioned by Reviewer TzHS), and
* Yang et al. [2023], Benton et al. [2020], van der Ouderaa and van der Wilk [2022], Immer et al. [2022], Kaba et al. [2023], Keller and Welling [2021], and Bouchacourt et al. [2021a], from our related work
to name just a few.

Unfortunately, it is impractical for us to go beyond these settings due to computational resource limitations.

Our set of experiments using dSprites, MNIST, and GalaxyMNIST, considering different types of symmetry transformations on each, demonstrate the general applicability of our method from small to large data sizes, from small to large dimensionalities, and for several different symmetries. The dSprites images, while simple, have a fairly large dimensionality (64 x 64 pixels) and are very plentiful (~740k images). GalaxyMNIST contains a small number of images (only 7k for training) of even larger dimensionality (64 x 64 x 3). This small data regime is perhaps more interesting since it demonstrates that our method is able to accurately capture the distribution over transformations without much data. Furthermore, we view the GalaxyMNIST images as ‘natural’ in that they come from real-world astronomy observations collected for the Galaxy Zoo DECaLS Campaign. Finally, these datasets demonstrate that our model can learn five affine transformations alone (MNIST, dSprites), three color transformations alone (MNIST), and both affine and color transformations together (GalaxyMNIST).

Like those excellent papers mentioned above, this paper has focused on novel ideas and methodological contributions rather than large-scale experiments. While we also value those experiments, we note that such engineering work often follows from work that builds up understanding and provides a promising proof-of-concept. Furthermore, we note that scaling up generative models is typically harder than their deterministic counterparts. Thus, we hope this will not be considered a major weakness of our work.

---

> ### Author Response · Authors · 2024-08-13
> **End of discussion period summary**
>
> Thank you to all of the reviewers for engaging with our rebuttal and the discussion period. We are very happy that two of the reviewers (TzHS and d33o) have raised their scores, that the other two reviewers (rf86 and GGon) maintain their recommendations to accept the paper, and that *the scores now lean towards an accept on average*.
>
> We'd like to highlight a few of the comments that came up during the discussion period:
> * "I regard this a **technically strong paper, with novel ideas and a good execution**, and therefore keep my recommendation for acceptance with score rating 8." – Reviewer rf86
> * "... **my main concern about demonstrating the correct conditional distribution learning [...] has been addressed**, I will substantially raise the score..." – Reviewer d33o
> * "**Most of my concerns have been addressed**... Overall, I maintain my opinion that this is a **well-written paper with clear motivation and reasonable experiment results**." – Reviewer GGon
>
> We believe that these comments highlight that the core concerns of the reviewers have been addressed, and that the reviewers see our paper as publication worthy due to it's novelty, technical strength, and strong presentation.
>
> The main remaining request is to include experiments on another (more complicated) dataset. While we haven't been able to do so due to the tight timelines and compute limitations in this discussion period, we will aim to include experiments based on the suggestion of reviewer TzHS in the camera ready version.

---

### Author Response · Authors · 2024-08-07
**Brief discussion of additional related work for reviewer TzHS**

*Group Equivariant Subsampling*. This paper introduces group-equivariant sub/upsampling layers. These layers can be used to construct group-equivariant CNNs with pooling-like layers. Our paper does not develop any novel architectures for group-equivariance, though our inference network could be made equivariant, as discussed in “Invariance of fω and the prototypes”. Thus, this line of work is related but orthogonal to ours.

*Learning Equivariances and Partial Equivariances from Data*. Like ours, this work focuses on learning partial symmetries from data. However, this work is more closely related to that of  Nalisnick and Smyth, [2018], van der Wilk et al. [2018], Benton et al. [2020], Schwöbel et al. [2021], van der Ouderaa and van der Wilk [2022], Rommel et al. [2022], Immer et al. [2022, 2023], Miao et al. [2023], and Mlodozeniec et al. [2023], in that it is concerned with the supervised learning regime.

*Deforming autoencoders: Unsupervised disentangling of shape and appearance*. Like ours, this paper focuses on unsupervised learning of prototypical objects (called ‘templates’ in their work) and transformations to convert such prototypes into observations. However, our work is more general, being applicable to a more general class of transformations. Conversely, theirs includes many interesting inductive biases specific to shape transformations. Their method is not probabilistic, meaning that they are unable to sample novel observations.

*Disentangling images with Lie group transformations and sparse coding*. Like ours, this paper learns to disentangle images into prototypes and their transformations. However, theirs differs from ours in two key ways. Firstly, they focus only on transformations corresponding to Lie groups whereas ours is more general. Secondly, they learn prototypes that are themselves composed of sparse combinations of learned dictionary elements. We see this work as complementary to ours, as the core idea of combining sparse coding with learned transformations could also be applied to our method.

---

### Decision · Program_Chairs · 2024-09-25

**Decision:**

Accept (poster)

**Comment:**

This is a well-written and well-motivated paper that presents a generative model of symmetry transformations. The main idea uses some recent work that treat relaxed symmetry as a latent variable over which inference can be performed. The experiments are thorough and well-described and validate the approach sufficiently. The reviews for the paper were mostly positive with one negative review. However, during the discussion period, the authors were partly able to address some of the reviewer concerns. On reading the paper, I tend to agree that the paper makes a good contribution and could be a valuable addition to the literature. The authors are requested to use the extra page to incorporate various points that have come up during the discussion phase.